# Methane efflux from an American bison herd

Paul C. Stoy[1-3]*, Adam A. Cook[4], John E. Dore[4,5], Natascha Kljun[6], William Kleindl[4], E. N. Jack Brookshire[4], Tobias Gerken[7,8]

[1]Department of Biological Systems Engineering, University of Wisconsin, Madison, WI, USA
[2]Department of Atmospheric and Oceanic Sciences, University of Wisconsin, Madison, WI, USA
[3]Department of Forest and Wildlife Ecology, University of Wisconsin, Madison, WI, USA
[4]Department of Land Resources and Environmental Sciences, Montana State University, Bozeman, MT, USA
[5]Montana Institute on Ecosystems, Montana State University, Bozeman, MT, USA
[6]Centre for Environmental and Climate Science, Lund University, Lund, Sweden
[7]Department of Meteorology and Atmospheric Science, The Pennsylvania State University, University Park, PA, USA
[8]School of Integrated Sciences, James Madison University, Harrisonburg, VA, USA

**Abstract.** American bison (*Bison bison* L.) have recovered from the brink of extinction over the past century. Bison reintroduction creates multiple environmental benefits but impacts on greenhouse gas emissions are poorly understood. Bison are thought to have produced some 2 Tg year$^{-1}$ of the estimated 9-15 Tg year$^{-1}$ of pre-industrial enteric methane emissions, but few measurements have been made due to their mobile grazing habits and safety issues associated with measuring non-domesticated animals. Here, we measure methane and carbon dioxide fluxes from a bison herd on an enclosed pasture during daytime periods in winter using eddy covariance. Methane emissions from the study area were negligible in the absence of bison (mean ± standard deviation = −0.0009 ± 0.008 μmol m$^{-2}$ s$^{-1}$) and were significantly greater than zero, 0.048 ± 0.082 μmol m$^{-2}$ s$^{-1}$, with a positively skewed distribution, when bison were present. We coupled bison location estimates from automated camera images with two independent flux footprint models to calculate a mean per-animal methane efflux of 58.5 μmol s$^{-1}$ bison$^{-1}$, similar to eddy covariance measurements of methane efflux from a cattle feedlot during winter. When we sum the observations over time with conservative uncertainty estimates we arrive at 81 g CH$_4$ bison$^{-1}$ day$^{-1}$ with 95% confidence intervals between 54 and 109 g CH$_4$ bison$^{-1}$ day$^{-1}$. Uncertainty was dominated by bison location estimates (46% of the total uncertainty), then the flux footprint model (33%) and the eddy covariance measurements (21%), suggesting that making higher-resolution animal location estimates is a logical starting point for decreasing total uncertainty. Annual measurements are ultimately necessary to determine the full greenhouse gas burden of bison grazing systems. Our observations highlight the need to compare greenhouse gas emissions from different ruminant grazing systems and demonstrate the potential for using eddy covariance to measure methane efflux from non-domesticated animals.

## 1 Introduction

The American bison (*Bison bison* L.) was hunted to near extinction during European expansion across North America (Flores 1991, Isenberg 2000, Smits 1995). Fewer than 100 reproductive individuals existed on private ranches in the United States during the late 19$^{th}$ Century from an original population of 30 – 60 million (Hedrick, 2009). The current bison population of

about 500,000 is due to the collective efforts of sovereign Indian tribes, government agencies, and private landowners (Gates et al., 2010; Sanderson et al., 2008; Zontek, 2007), all of whom have spurred a growing interest in bison reintroduction. The bison population is likely to further increase, increasing the incentive for researchers and land managers to understand the environmental impacts of their expansion.

The ecological role of bison has become better understood as populations have recovered (Allred et al., 2001; Hanson 1994; Knapp et al., 1999). Bison feed preferentially on grasses (Plumb and Dodd, 1993; Steuter and Hidinger, 1999) and often enhance forb diversity as a result (Collins, 1998; Hartnett et al., 1996, Towne et al., 2005). They tend to graze in preferred meadows during winter and search broadly for the most energy-dense forages during the growing season (Fortin et al., 2003 Geremia et al., 2019), often in areas which have recently burned (Allred et al., 1991; Coppedge and Shaw, 1998; Vinton et al., 1993). Combined, these observations suggest that bison select for forage quality rather than quantity which likely impacts their efflux of methane – which all ruminants emit – because ruminant methane emission is related to feed quality (Hammond et al., 2016) including cellulose and hemicellulose intake (Moe and Tyrrell, 1979). It remains unclear how much methane results from the cellulose-rich grass-dominated diet of bison given their preference for fresh foliage, and if management for bison may increase or diminish the greenhouse gas burden of ruminant-based agriculture.

Atmospheric methane concentrations have been rising at an accelerated rate since 2016 for reasons that remain unclear (Nisbet et al., 2019) and there is an urgent need to improve our understanding of its surface-atmosphere flux. Between 30 and 40 percent of current anthropogenic methane emissions are due to enteric fermentation in livestock (Kirschke et al., 2013) and the greenhouse gas burden of cattle alone is some 5 Pg of carbon dioxide equivalent per year (Gerber et al., 2013; FAO, 2017). Methane emission estimates from livestock have tended to increase as more information becomes available (Beauchemin et al., 2008; Thornton and Herrero, 2010; Wolf et al., 2017), further emphasizing their critical role in global greenhouse gas budgets (Reisinger and Clark, 2017). Reducing unnecessary greenhouse gas emissions is a global imperative for Earth system management and reducing enteric methane sources is seen as a promising approach to do so (Boadi et al., 2002; DeRamus et al., 2003; Herrero, et al., 2016; Hristov et al., 2013; Johnson and Johnson, 1995; Moss et al., 2000).

Bison in North America are thought to have been responsible for some 2.2 Tg year$^{-1}$ (Kelliher and Clark, 2010; Smith et al., 2016) of the 9-15 Tg year$^{-1}$ of pre-industrial enteric methane emissions (Thompson et al., 1993; Chappellaz et al., 1993; Subak, 1994). Enteric $CH_4$ emissions from wild ruminants in the United States in the pre-settlement period comprised nearly 90% of current $CH_4$ emissions from domesticated ruminants assuming an historic bison population size of 50 million (Hristov, 2012), further demonstrating the importance of bison to methane fluxes in the past. The current and future contribution of non-domesticated ungulates to methane fluxes are uncertain (Crutzen et al., 1985). Previous approaches used inventory approaches or scaling equations that were not derived using methane efflux measurements from bison; the only direct bison methane flux observations that we are aware of measured 30 L of methane per kg dry food intake (17 g methane per kg dry food intake) from one-year-old penned female bison fed alfalfa pellets (Galbraith et al., 1998), more than elk (*Cervus elaphus*) and white-tailed deer (*Odocoileus virginianus*) on a dry matter intake basis and similar to dairy cattle fed high maize silage (Hammond et al., 2016). Cattle methane emissions tend to be greater when fed alfalfa than grass (Chaves et al., 2006) such that existing

published values may not represent an accurate estimate of the methane efflux from bison in a natural field setting, which has not been measured to date.

Here, we measure methane flux from a bison herd on winter pasture using the eddy covariance technique (Dengel et al., 2011; Felber et al., 2015; Prajapati and Santos, 2018; Sun et al., 2015). We use flux footprint analyses combined with bison locations determined using automated cameras to estimate methane flux on a per-animal basis and discuss observations in the context of eddy covariance methane flux measurements from other ruminants.

## 2 Methods

### 2.1 Study site

The study site is a 5.5-hectare fenced pasture on the Flying D Ranch near Gallatin Gateway, Montana, USA (45.557, −111.229) on a floodplain immediately west of the Gallatin River (Figure 1). Daily high temperatures average 1.6 °C and daily low temperatures average −11.5 °C at Bozeman Yellowstone International Airport (BZN), located 24 km north-northeast of the site, during the November – February measurement period. BZN records an average of 18.2 mm of precipitation per month

during November – February, almost entirely as snowfall. A herd of 39 bison entered the pasture on November 17, 2017 and left on February 3, 2018. The mean (standard error) bison weight measured by the landowners on November 16, 2017 before bison entered the pasture was $329 \pm 28$ kg and the bison varied in age from 0.5 to 7.5 years old (Table S1). Bison consumed a mixture of perennial grasses grown *in situ* that was supplemented by perennial grass hay grown in nearby fields (Table S2) delivered every three days on average (Table S3) such that the management approach shares features with pasture and feedlot

systems.

### 2.2 Instrumentation

A 3-m tower was installed near the center of the study pasture during November 2017 (Figure 1) and surrounded by electric fencing to avoid bison damage. Four game cameras (TimelapseCam, Wingscapes, EBSCO Industries, Inc., Birmingham, AL, USA) were mounted to the tower and pointed in cardinal directions. Two additional game cameras were mounted near the

90 pasture edge facing the tower. Cameras captured images every five minutes and an example of an individual image from the south-facing camera located on the northern edge of the study pasture is shown in Figure 2. Bison locations at the half-hourly time interval of the eddy covariance measurements were estimated by manually attributing bison locations to squares in a 20 m grid overlaid on the pasture area (Figure 1). The 20 m grid size represents the grid that we felt that we were able to attribute bison locations given features of the field that could be identified by camera, and we treat these observations as an initial guess

that is subject to uncertainty. We test the sensitivity of per-animal methane efflux estimates to bison location estimates as described in the *Spatial Uncertainty* section below.

Incident and outgoing shortwave and longwave radiation and thereby the net radiation were measured using a NR01 net radiometer (Hukseflux, Delft, The Netherlands) mounted 1.5 meters above ground level. A SR50 sonic distance sensor (Campbell Scientific Inc., Logan, UT, USA) was installed at 1.3 m to gauge snow depth, and air temperature and relative humidity were measured at 2.25 meters using a HMP45C probe (Vaisala, Vantaa, Finland). Average 0–30 cm soil moisture and temperature were collected using CS650 probes (Campbell Scientific). Meteorological variables were measured once per minute, and half-hourly averages were stored using a CR3000 datalogger (Campbell Scientific).

Three-dimensional wind velocity was measured using a CSAT-3 sonic anemometer (Campbell Scientific) at 2.0 m above the ground surface. Carbon dioxide mixing ratios were measured at 10 Hz using a LI-7200 closed-path infrared gas analyzer (LI-COR Biosciences, Inc.) with inlet placed at the same height as the center of the sonic anemometer. Methane mixing ratios were measured at 10 Hz using a LI-7700 open-path infrared gas analyzer (LI-COR Biosciences, Inc., Lincoln, NE, USA) with the center of the instrument likewise located at 2.0 m and a 22 cm horizontal offset from the sonic anemometer; open- and closed-path infrared gas analyzers for eddy covariance have similar performance in field settings (Detto et al., 2011; Deventer et al., 2019). We use the atmospheric convention in which flux from biosphere to atmosphere is positive. Measurements were made during winter daytime hours from 0700 to 1700 local time to avoid depleting the battery bank and to ensure sufficient light to estimate bison location using game cameras. Flux measurements began on November 14, 2017 and ended on February 14, 2018.

Bison are dangerous and will charge humans. Their presence complicated data retrieval and game camera upkeep; some high-frequency flux measurements were overwritten and cameras shut down during exceptionally cold periods, resulting in missing measurements. Simultaneous flux and photographic data were obtained for the January 7, 2018 to February 13, 2018 period excluding January 10, 2018 when instruments were obstructed by snowfall. Flux data without accompanying game camera footage were obtained for the periods from November 14 through 29, 2017 and December 31, 2017, through January 6, 2018.

### 2.3 Flux calculations

Methane and carbon dioxide fluxes were calculated using EddyPro (LI-COR Biosciences, Lincoln, NE, USA). Standard double rotation, block averaging, and covariance maximization with default processing options were applied. Spike removal was performed as described by Vickers and Mahrt (1997) and spikes were defined as more than 3.5 standard deviations from the mean mixing ratio for carbon dioxide and more than 8 standard deviations from the mean mixing ratio for methane given the expectation of intermittent methane spikes from the bison herd. The default drop-out, absolute limit, and discontinuity tests were applied using the default settings following recommendations by Dumortier et al. (2019), and the Moncrieff et al. (1997) and Moncrieff et al. (2004) low- and high-pass filters were applied. The Webb-Pearman-Leuning correction (Webb et al., 1980) was applied to calculate methane efflux using the open-path LI-7700 sensor. Estimates of storage flux in the 2 m airspace below the infrared gas analyzers were assumed to be minor and excluded from the flux calculation. Flux measurements for which the quality control flag was greater than 1 following Mauder and Foken (2004) (see also Foken et al., 2004) were

discarded and the net effect of all corrections when bison were present was a methane flux reduction of 14%. Measurements that exceeded an absolute value of 1 μmol m$^{-2}$ s$^{-1}$ for the case of methane flux and 20 μmol m$^{-2}$ s$^{-1}$ for the case of carbon dioxide flux were discarded following an analysis of the probability distribution of observations. We tested the sensitivity of flux measurements to the friction velocity ($u*$) to see if measurements made under conditions of insufficient turbulence should be excluded from the analysis despite the daytime-only flux measurement approach.

## 2.4 Flux footprint modelling

The eddy covariance flux footprint was calculated using the approach of Hsieh et al. (2000) extended to two dimensions following Detto and Katul (2006). Such analytical footprint models have been found to give minimally biased estimates of point-source fluxes in field settings (Dumortier et al., 2019). We performed the footprint analysis on a 1-m grid and aggregated values to the 20 × 20 m grid to which the bison locations were estimated (Figure 1). To further characterize the uncertainty in our per-animal methane flux estimates, described next, we also applied the flux footprint parameterization method of Kljun et al. (2015) aggregated to the same 20 × 20 m grid. The Kljun et al. (2015) model performed best in point-source experiments (Heidbach et al., 2017) and is widely used by the flux community. Figure 3 demonstrates an example of flux footprints for both models for a single half-hourly period.

The momentum roughness height ($z_{0m}$) is required by both footprint models. Instead of assuming a constant $z_{0m}$ over snow of 0.001 m (Andreas et al., 2004), we followed the approach of Baum et al. (2008) who calculated a unique $z_{0m}$ for each half-hour eddy covariance measurement for a cattle feedlot system by rearranging the wind profile equation:

$$z_{0m} = \frac{z - d}{exp(ku/u_* + \psi_m)} \tag{1}$$

Where $z$ is measurement height, $u$ is wind speed, $k$ is the von Karman constant, and $\psi_m$ is the correction factor for atmospheric stability, here following Brutsaert (1982). The zero-plane displacement ($d$) for a field with obstacles is calculated following Verhoef et al. (1997):

$$d = z - \frac{z(1-\exp(-\sqrt{42a})}{\sqrt{42a}}. \tag{2}$$

where $a$ is the frontal area index of the obstacles (Raupach, 1994), here bison:

$$a = \frac{nbh}{S}. \tag{3}$$

The calculation of $a$ uses the number of animals ($n = 39$), the size of the pasture ($S$, m$^2$), and the average breadth ($b$, m) and height ($h$, m) of the animals. We used established relationships for beef cattle as a function of weight (ASABE, 2006) given the lack of similar equations for bison. $h$ was adjusted upward by 50% such that the height of adult males better-matched average values of fully-grown bison on the order of 1.8 m. The methane source location was assumed to be near the ground or snow surface per the typical posture of bison assuming that most methane efflux in ruminants is from erucation. We used the mean value of per-animal flux estimated by the two footprint models and the variance between them to calculate footprint uncertainty.

## 2.5 Per-bison methane flux estimation

Given that mean methane emissions were not significantly different from zero in the absence of bison – as detailed in *Results* – we assume that observed methane emissions are due to bison in the flux footprint. The relative contribution of bison to each
half-hourly eddy covariance measurement was calculated by expanding the approach of Dumortier et al. (2019) (see also Prajapati and Santos (2019)) for multiple point sources. From the definition of the footprint function (e.g. Schmid, 1997), the measured density of a scalar $X$, $F_X$, for our study area of $8 \times 12$ grid cells (Figure 1) is:

$$F_X = \sum_{i=1}^{8} \sum_{j=1}^{12} F_{ij} \phi_{ij} \Delta x_{ij} \Delta y_{ij} \tag{4}$$

where $\phi_{ij}$ is the value of the footprint function in grid cell $ij$, $x$ and $y$ are the dimensions of the 20 m grid cells (i.e. 400 m$^2$), and $F_{ij}$ is the flux from grid cell $ij$. We have $n = 39$ sources (i.e. bison) that are free to wander to any grid cell $ij$, and we have
no basis for identifying individual bison given the resolution of the cameras, noting that this is possible using higher-resolution cameras (Merkle and Fortin, 2013) or GPS instruments. We also have no basis for determining if the methane sources of individual bison are different using our approach, so we must assume that methane efflux from each bison is equal, i.e.

$$F_{ij} = n_{ij} \langle f_{ij} \rangle \tag{5}$$

Where $n_{ij}$ is the number of bison in grid cell $ij$ (i.e. per 400 m$^{-2}$), $\langle f_{ij} \rangle$ is the average flux per bison in grid cell $ij$ and the average per-bison flux $\langle f_x \rangle$ is:

$$\langle f_x \rangle = \frac{F_X}{\sum_{i=1}^{8} \sum_{j=1}^{12} n_{ij} \phi_{ij}} \tag{6}$$

We only adopt this approach for calculating average methane efflux per bison as measured carbon dioxide fluxes in the absence of bison were significantly greater than zero. Methane efflux values less than $-200$ µmol bison$^{-1}$ s$^{-1}$ and greater than 300 µmol bison$^{-1}$ s$^{-1}$ were treated as outliers and excluded based on an analysis of the probability distribution of observations. After filtering for eddy covariance measurement quality, outliers, and photograph availability, measurements with bison in the flux footprint were available on 158 half hours when applying the Hsieh et al. (2000) footprint model and 146 half hours when
applying the Kljun et al. (2015) footprint model, noting that their dimensions differ (e.g. Fig. 3).

## 2.6 Uncertainty estimation

Our observations are subject to multiple sources of uncertainty including uncertainty from eddy covariance measurements, footprint models, and bison location estimates. Uncertainty of the eddy covariance methane flux measurements was determined by Deventer et al. (2019) to be between $6 - 41\%$ for half-hourly fluxes. We use an uncertainty of 41% as we are primarily
concerned with providing a conservative uncertainty assessment and take the absolute value of the measurements multiplied by this percentage to calculate uncertainty due to eddy covariance measurements. Uncertainty due to the flux footprint was

calculated as the mean percent difference in per-bison flux calculated using the Hsieh et al. (2000) and Kljun et al. (2015) footprint models.

Uncertainty due to bison location estimates is more difficult to calculate. The location of bison in the pasture was approximated visually by identifying the position of bison in relation to static cues in the study area using five-minute photographs. Observations were then aggregated to half-hourly flux measurement periods. This approach results in spatial uncertainty in bison location, especially due to movements within half-hourly periods and potential misallocation to nearby grid cells (Figure 1). We acknowledge that uncertainty in bison location estimate is likely using our approach and explored the sensitivity of per-bison methane flux estimates to bison location using stochastic simulations in order to arrive at a conservative uncertainty estimate.

The camera measurements resulted in many pixels where bison were not observed (e.g. Figure S1), but there is a finite probability that this absence was in error. Pixels near populated pixels likely have a higher probability that bison were located within them because small movements within half-hour periods were common and because their locations may have been misallocated due to measurement uncertainty. We therefore sought an approach that simulates a spatial distribution of bison that is constrained by the camera measurements. To do so, we treated the camera measurements as an initial guess of their location that helped us define a likelihood surface. The likelihood surface was determined using two-dimensional Tikhonov Regularization (Tikhonov and Arsenin, 1977), a classic mathematical technique to solve ill-posed problems, here the challenge of estimating the likelihood of bison location with intermittent and uncertain observations as described in detail in the Supplemental Information. The probability of the 39 bison landing in a pixel is informed by this likelihood surface, and we used 100 simulations for both the Hsieh et al. (2000) footprint and the Kljun et al. (2015) footprint along four different values of the spatial smoothness of the probability surface defined by the Lagrange multiplier (Equation S1). An example of a likelihood surface generated for a single half-hour observation of bison locations and different values of the Lagrange multiplier is shown in Figure S1. We explore the sensitivity of per-bison methane emissions to the Tikhonov Regularization approach in the Supplemental Information (Figures S2 and S3).

We took the percent difference between the calculated per-bison methane emissions and values from the 200 stochastic simulations as the uncertainty due to bison location. Total uncertainty was then calculated by summing variances for the spatial uncertainty, footprint model uncertainty, and eddy covariance uncertainty. We suggest strategies for reducing uncertainty in the *Discussion* section.

## 3. Results

### 3.1 Meteorology

Air temperature averaged −2.8 °C and soil temperature averaged −0.3 °C during the measurement period (Figure 4A). Incident shortwave radiation ranged between 100 and 400 W m$^{-2}$ during peak daylight hours (1000-1400 hours local time) across the study period, and clear conditions were common except for four weeks beginning in mid-December (Figure 4B). Snow depth within the tower enclosure increased from 0.15 m to nearly 0.4 m in late 2017 and decreased to 0.1 m beginning in late January

2018 (Figure 4C) noting that snow outside of the electrified tower enclosure was often trampled (see Figure 2). The mean (median) wind direction was 221° (208°) during periods when visible imagery of bison locations was available and eddy covariance measurements passed quality control checks (Figure 5).

## 3.2 Gas flux

Half-hourly methane fluxes averaged $0.048 \pm 0.081$ μmol m$^{-2}$ s$^{-1}$ (mean ± standard deviation) and carbon dioxide fluxes averaged $1.6 \pm 1.4$ μmol m$^{-2}$ s$^{-1}$ when bison were present (Figure 6), noting again that measurements were made only during daytime periods. Median $z_{0m}$ was 0.017 m in the absence of bison and 0.028 m when bison were present, the latter similar to $z_{0m}$ established for grass fields with intermittent obstacles (Wieringa, 1992). Methane flux in the absence of bison averaged $-0.0009 \pm 0.008$ μmol m$^{-2}$ s$^{-1}$ and carbon dioxide flux averaged $0.64 \pm 1.0$ μmol m$^{-2}$ s$^{-1}$, significantly lower than when bison

were present ($P < 0.001$ for both $CH_4$ and $CO_2$). $CO_2$ flux was significantly related to methane flux and explained 52% of its variance when bison were present but only 7% when they were absent (Figure 7). $CO_2$ flux was significantly and positively related to air and soil temperature across the entire measurement record ($P < 0.001$ in both cases), but methane flux was not. There were no significant temporal patterns of methane flux during the daytime periods investigated here, and neither incident nor net radiation were related to methane flux. Methane flux was not significantly different during days when feed was

delivered ($0.051 \pm 0.083$ μmol m$^{-2}$ s$^{-1}$) and days when it was not ($0.035 \pm 0.10$ μmol m$^{-2}$ s$^{-1}$) ($P = 0.075$) when bison were present.

Methane flux was significantly and positively related to friction velocity in the absence of bison at $u^*$ values greater than 0.2 m s$^{-1}$ ($P = 0.003$) but not positively related to $u^*$ values less than 0.2 m s$^{-1}$, indicating that flux measurements were unrelated to friction velocity values commonly associated with insufficient turbulence (Figure 8A). Carbon dioxide flux was not related

to $u^*$ in the absence of bison (Figure 8B) but negative values were observed at $u^*$ values greater than 0.45 m s$^{-1}$. Given these observations, we did not apply a $u^*$ filter to our eddy covariance measurements, which were made only during daytime periods. We discuss potential reasons for the observed increase in methane flux and negative $CO_2$ flux with high values of $u^*$ in the *Discussion* section.

## 3.3 Bison location and methane efflux

Timelapse camera footage yielded usable imagery for 444 half-hourly periods of which 245 half-hourly periods had available eddy covariance observations and of which 177 had eddy covariance measurements that passed quality control criteria. Bison tended to aggregate in an area on the west side of the pasture near the location where supplemental hay was often provided (Figure 9A). They intermittently visited the area north of the tower in mornings and afternoons and intermittently made sporadic mass movements to the southernmost edge of the field near its gate during midday periods (Figure 9B-D).

Bison were located within the 90% flux footprint 40% of the time (Figure 10). There were 158 half-hourly observations with bison in the flux footprint when applying the Hsieh et al. (2000) footprint model and 146 observations were available when

applying the Kljun et al. (2015) footprint model and an average of eight (seven) bison within the 90% flux footprint of the Hsieh et al. (2000) (Kljun et al. (2015)) models. When excluding periods for which bison were absent from the flux footprint, this value increased to 21 (20), respectively (Figure 10). Per-bison methane emission estimates when using the Hsieh et al. (2000) footprint model had a mean (± standard error) of $55 \pm 0.96$ µmol bison$^{-1}$ s$^{-1}$ and a median of 29 µmol bison$^{-1}$ s$^{-1}$ as a result of the positively skewed measurement distribution (Figure 11A). These estimates are 11% lower than per-bison methane emission estimates from the Kljun et al. (2015) footprint model, which returned a mean (± standard error) of $62 \pm 0.91$ µmol bison$^{-1}$ s$^{-1}$, which demonstrates that per-animal flux estimates are sensitive to flux footprint methodology.

Per-bison methane flux estimates from stochastic simulations of bison location were sensitive to the smoothness of the likelihood surface (Figure 12). Combining per-bison methane flux estimates from all 100 simulations resulted in a standard deviation of 6.2 µmol bison$^{-1}$ s$^{-1}$ when using the Hsieh et al. (2000) model and 5.8 µmol bison$^{-1}$ s$^{-1}$ when using the Kljun et al. (2015) model. If we sum variances to combine uncertainties due to spatial uncertainty, flux footprint, and the eddy covariance measurements themselves and extrapolate observations to the daily time scale commonly used in other methane flux studies, we arrive at a mean daily methane efflux of 81 g $CH_4$ bison$^{-1}$ day$^{-1}$ with 95% confidence intervals between 54 and 109 g $CH_4$ bison$^{-1}$ day$^{-1}$. The uncertainty is dominated by uncertainty due to bison location (46% of the total uncertainty), then the flux footprint model (33%), then the eddy covariance measurements (21%).

## 4 Discussion

The eddy covariance flux footprint analysis coupled to bison location estimates from automated camera images resulted in a mean methane flux of 55 µmol bison$^{-1}$ s$^{-1}$ when applying the Hsieh et al. (2000) footprint model and 62 µmol bison$^{-1}$ s$^{-1}$ when applying the Kljun et al. (2015) footprint model for a combined mean ± variance of 58.5 µmol bison$^{-1}$ s$^{-1}$, or 81 g $CH_4$ bison$^{-1}$ day$^{-1}$ with 95% confidence intervals between 54 and 109 g $CH_4$ bison$^{-1}$ day$^{-1}$. Measurements were made during daytime periods in winter and are sensitive to estimates of bison location (Figure 12). If we naively assume that methane flux from bison varies negligibly across the full diurnal and seasonal range, a notion that needs to be substantiated, our measurements roughly correspond to 30 kilograms of methane per bison per year with 95% confidence intervals between 20 and 40 kilograms of methane per bison per year. Below, we discuss potential reasons for the bison methane emissions observed here as well as strategies for reducing uncertainty in eddy covariance measurements of methane flux from non-domesticated ruminants.

### 4.1 Bison methane flux observations in the context of other grazing systems

It is important to study methane emissions from other grazing systems to place our observations into a broader context and, moving forward, to design grazing systems that minimize greenhouse gas burdens. From this perspective, our simple seasonal scaling exercise may underestimate or overestimate methane emissions from bison grazing systems for multiple reasons that must be kept in mind when interpreting results. Methane emissions from cattle have been observed to be on the order of 10-17% higher in summer than winter (Todd et al., 2014; Prajapati and Santos, 2018; Prajapati and Santos, 2019) such that our

wintertime methane flux observations may be lower than what full annual measurements would yield. Our observations were similar to wintertime measurements of beef cattle in a feedlot, on the order of 75 g $CH_4$ animal$^{-1}$ day$^{-1}$ (Prajapati and Santos, 2019), which to a first order suggests that bison and cattle grazing systems may have similar methane efflux. Our study pasture shares features with both feedlot and grazing systems with important implications for methane efflux. The bison were free to graze (Fig. 2) but were also supplied supplemental hay at regular intervals (Table S2). In other words, it is safe to assume that the animals were well-fed, which cannot be assumed to be the case during winter in a wildland bison grazing system. Cattle in Africa were observed to have higher methane yields per unit feed when feed intake was below maintenance levels during the dry season when food is scarce (Goopy et al., 2020). Bison in natural grazing systems may also have a greater methane yield per unit feed when food is scarce during winter, but lower total methane efflux if less feed is consumed given the strong relationship between feed intake and methane production (Johnson & Johnson, 1995).

We did not observe significant differences in methane efflux over the course of the day noting that our observations were limited to daytime periods because we had little basis to determine animal location at night. Other studies have observed higher methane efflux from cattle during feeding times (Gao et al., 2011), but bison also frequently graze at night, leaving it unclear if they also exhibit daytime and nighttime differences in methane flux with implications for scaling flux across time. Methane efflux was not significantly higher during days when supplemental hay was provided ($P = 0.075$), suggesting that the opportunity to graze throughout the day regardless of supplemental food muted any diurnal methane efflux cycle that may have been present if they fed at preferred times.

Nutritional needs also impact methane efflux; dairying buffalo cows for example are estimated to have higher methane emissions than other buffalo (Cóndor et al. 2008). The study herd comprised numerous pregnant females (Table S1) that have higher metabolic requirements such that methane flux values may be higher than a herd with fewer pregnant animals. Taken as a whole, there is no evidence from our measurements that bison have more or less methane efflux than typical values reported for cattle. We note that it is critical to make full year-round methane flux measurements to understand the seasonal course of bison methane efflux to establish defensible annual sums.

**4.2 Methane and carbon dioxide efflux in response to environmental variables and bison presence**

Methane flux was not related to air or soil temperature but was related to $u^*$ – especially at relatively high values of $u^*$ – in the absence of bison (Figure 8). These observations are consistent with a potential pressure pumping mechanism for trace gases through snow at higher wind speeds (Bowling and Massman, 2011) although it is unclear why this relationship exists for methane flux and not carbon dioxide flux as is frequently found in snow-covered conditions (Rains et al., 2016). Carbon dioxide flux at high values of $u^*$ was negative indicating net $CO_2$ uptake by the biosphere, which is unlikely in our study site

during winter, suggesting that values with excessively high $u*$ may need to be filtered, but with only five observations of $CO_2$ flux less than zero it is unclear how to apply such a filter in our case.

Insufficient evidence exists in our data record to attribute observed methane efflux to the onset of freezing conditions in soil (Mastepanov et al., 2008). We note that extensive snow trampling (e.g., Figure 2) likely resulted in a situation where snow depth (Figure 4C) and its insulating effect on soil temperature (Figure 4A) varied across the field and therefore differed from snow and soil measurements taken within the instrumentation enclosure. Regardless, mean methane flux when bison were absent, $-0.0009$ μmol m$^{-2}$ s$^{-1}$, was nearly two orders of magnitude less than the mean methane flux when bison were present,

0.041 μmol m$^{-2}$ s$^{-1}$. Whereas we cannot exclude – and in fact expect – non-zero background methane fluxes from non-bison sources in a grassland in winter in the vicinity of a riparian area (Figure 1, Merbold et al., 2013; McLain and Martens, 2006; Mosier et al., 1991), these are minor compared to the $CH_4$ flux attributable to bison (Figures 6 and 7). Bison are associated with a distinct methane flux signature as shown by the immediate decline of methane fluxes following their removal from the study pasture (Figure 6) and strong relationship with carbon dioxide flux (Figure 7) given the common source of respiration

and most enteric methane losses from the muzzles of ungulates. Methane flux was related to carbon dioxide flux when bison were present or absent (Figure 7), suggesting both soil and ruminant sources (and in the case of methane sinks) of both gases (Baldocchi et al., 2012; Gourlez de la Motte et al., 2019).

It is important to note that potential methane fluxes from bison manure may have been dampened by freezing conditions but may be an important methane source during warmer conditions if it enters anoxic conditions. Manure is thought to contribute

a nontrivial portion (10-14 Tg $CH_4$ yr$^{-1}$) of total global ruminant methane efflux (77 Tg $CH_4$ yr$^{-1}$, Johnson and Ward 1996; Moss et al., 2000) noting that some farm-scales studies arrive at lower percentages (Taylor et al., 2017). Though we did not observe higher methane efflux early in the study period when soil temperature was above freezing nor temperature sensitivity of methane efflux in the presence or absence of bison, it is important to note that field-scale methane efflux may be diminished by the thermal environment of manure in our measurements but is still likely to be relatively low in a rangeland setting (Steed

Jr. and Hashimoto, 1994).

## 4.2 Bison spatial distribution and measurement uncertainty

Ruminant behavior is an important consideration when measuring field-scale efflux (Gourlez de la Motte et al., 2019). The spatial distribution of bison in the study pasture often varied from morning to midday and afternoon (Figure 9). It is difficult to infer from the available data whether the study bison are more active during morning and evening hours in the pasture

environment like cattle (Gregorini 2012). Supplemental hay was made available to the bison approximately 50 meters west of the tower and increases in the frequency of bison appearance there are associated with the animals' preferred feeding times after dawn and before dusk, but observed methane flux did not vary as a function of time of day (e.g. Dengel et al. 2011) as noted above. Regardless, ruminant methane flux measurements are simpler to make when animals congregate (Coates et al., 2017; Tallec et al., 2012) as was often observed in our study (e.g. Figures 2, 9 & 10). Aggregation behavior in our study bison

herd was often upwind of the eddy covariance tower (Figures 5 & 9) and resulted in more overlap between flux footprint and bison location than would have occurred if bison locations were randomly distributed throughout the study area, emphasizing the importance of tower placement in eddy covariance studies of grazing systems.

Despite the largely favorable location of the herd with reference to wind direction and the flux footprint, spatial uncertainties in bison location dominated the total uncertainty calculated here. More accurate location observations are a logical way to
reduce this uncertainty. Uncertainties in flux footprint modelling for methane source attribution were also non-trivial on the order of 33% of total uncertainty. Footprint models of the type used here have been found to accurately estimate point sources of trace gas flux (Heidbach et al., 2017; Dumortier et al., 2019), but it is important to note that footprint modelling techniques play a large role in the spatial attribution of observed fluxes of ruminant trace gas flux (Felber et al., 2015). Prajapati and Santos (2018), for instance, found that an analytical model (Kormann and Meixner 2001) predicted flux footprint areas five to
six times larger than did an approximation of a Lagrangian dispersion model (Kljun et al., 2002) did, such that footprint model uncertainty is a major source of uncertainty for measuring methane flux from multiple point sources as we also find here. Regarding the footprint model it is also important to note that emitted gas is warmer than the surrounding environment in our case. It is unclear how well typical eddy covariance flux footprint models simulate the release location of heated parcels but we note that heat is also transferred more efficiently than passive scalars like methane in the convective sublayer (Katul et al.,
1995) such that methane transport should not be assumed to behave like heat. It is also unclear for our case if a point near the snow surface accurately represented the typical parcel release height. We were unable to track individual animals with different muzzle heights, noting that the animals were also frequently grazing with muzzle below the snow surface such that the true parcel release point represented a wide range of heights that we had little basis to simulate from available observations.

### 4.3 Future directions for greenhouse gas accounting in ruminant grazing systems

Methane efflux cannot be completely removed from ruminant grazing systems; some 4.6 – 6.2% of gross energy intake is lost as methane in cattle, sheep and goats worldwide (Johnson and Ward 1996) with cattle often falling on the higher end of the observed range (Lassey et al., 1997). But there are other aspects of bison ecology that merit consideration when designing greenhouse gas-cognizant grazing systems. For example, cattle tend to graze close to water more frequently than bison do (Allred et al., 2011) with unclear consequences for riparian vegetation, water quality, and potential methane efflux from
wallows. Cattle also tend to graze for longer periods than bison (Plumb and Dodd, 1993) and it is unclear if there is an associated consequence for methane efflux. Future work should consider the large inter-animal variability in methane efflux (Lassey et al., 1997), possibly using advanced techniques for identifying individual animals through photographs (Merkle and Fortin, 2013) or tracking devices (Felber et al., 2015). Animal age and size are also important factors in ruminant methane efflux (Jiao et al., 2014) and individual tracking may improve our estimates of this variability in a field setting. That being
said, it will be difficult to measure the methane contributions of different animals in species that tend to herd using eddy covariance.

Adding seasonal foraging behavior, estimating emissions from individual animals, and addressing seasonal and inter-annual variability and trends in forage nutrition are likely to further improve prediction of methane emissions from grazing systems (Moraes et al., 2013). Advanced eddy covariance algorithms are also likely to improve flux estimates on short time scales noting that non-stationary bursts have not been found to create systematic bias in methane budgets measured over longer time periods using eddy covariance (Göckede et al., 2019). Of these, advanced footprint attribution techniques like Environmental Response Functions designed to create improved maps of surface-atmosphere fluxes (Metzger et al., 2013; Xu et al., 2017) may be uniquely applicable to the challenging case presented by grazing systems with mobile point sources and intermittent biogeochemical hotspots created by animal waste. Going forward, increases in atmospheric carbon dioxide concentrations are likely to decrease forage quality (Jégo et al., 2013), resulting in higher leaf carbon to nitrogen ratios and increasing ruminant methane emissions (Lee et al., 2017), all else being equal. Understanding greenhouse gas fluxes from ruminants is therefore likely to be even more important in the future. An ongoing interest in bison reintroduction and ungulate ecology coupled with established micrometeorological measurement techniques will help us understand the present and future role that bison and other alternative grazing systems play in the Earth system.

## 4.4 Conclusions

We measured methane efflux from a bison herd from an enclosed pasture using the eddy covariance method. Measurements were made during winter and background methane flux measurements in the absence of bison were not different from zero. Bison were free to graze and were also fed supplemental hay, which likely resulted in different methane efflux from that of a natural herd. Regardless of potential differences in greenhouse gas fluxes between wild and managed bison, bison are not domesticated and it is difficult to make measurements of their greenhouse gas efflux using standard techniques like chambers or the sulfur hexafluoride method. Our results suggest that eddy covariance is a promising method for measuring trace gas fluxes from non-domesticated ruminants and that improved technologies for tracking animal movement is a logical way to reduce total uncertainties in observations. There is little evidence from our observations that methane efflux from the study herd differed from wintertime methane efflux from a cattle feedlot system, but full annual flux observations are necessary to understand if methane efflux from bison differs from the cattle management systems that originally replaced them, and are now in turn being increasingly replaced by bison across much of their native range by a diverse group of Native American Tribes and private landowners who share a common interest in bison reintroduction and conservation.

**Acknowledgements**

We wish to acknowledge the Indigenous Nations on whose ancestral lands the study took place and recognize that the infrastructure used for this project was built on Indigenous land. We recognize multiple Indigenous Nations as past, present, and future caretakers of this land whose stewardship of the region was interrupted through their physical removal by the 1830 Indian Removal Act and through US Assimilation policies explicitly designed to eradicate Indigenous language and ways of

being until the 1970s. PCS acknowledges support from the U.S. National Science Foundation awards DEB-1552976 and OIA-
1632810, the USDA National Institute of Food and Agriculture Hatch project 228396, the Multi-State project W3188, the
Graduate School at Montana State University, and the University of Wisconsin – Madison. Funding for the LI-7700 methane
analyzer used in this work was provided to JED by NSF-EPSCoR award EPS-1101342 and Montana State University. Daniel
Salinas, Gabriel Bromley, Zheng Fu, and James Irvine provided technical assistance and Aaron Bird Bear and colleagues
provided cultural knowlege. This work could not have been completed without permission of Turner Enterprises, Inc. and the
assistance of Carter Kruse and Danny Johnson.

**Code/Data availability**

Eddy covariance and micrometeorological data have been submitted to Ameriflux for publication at
https://ameriflux.lbl.gov/sites/siteinfo/US-Tur.

**Author contributions**

PCS designed the study with AC, JD, and WK and wrote the manuscript with all coauthors. AC collected data and analyzed it
with PCS and TG. NK assisted with the footprint analysis.

**Competing interests**

The authors declare no competing interests.

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

**Figures**

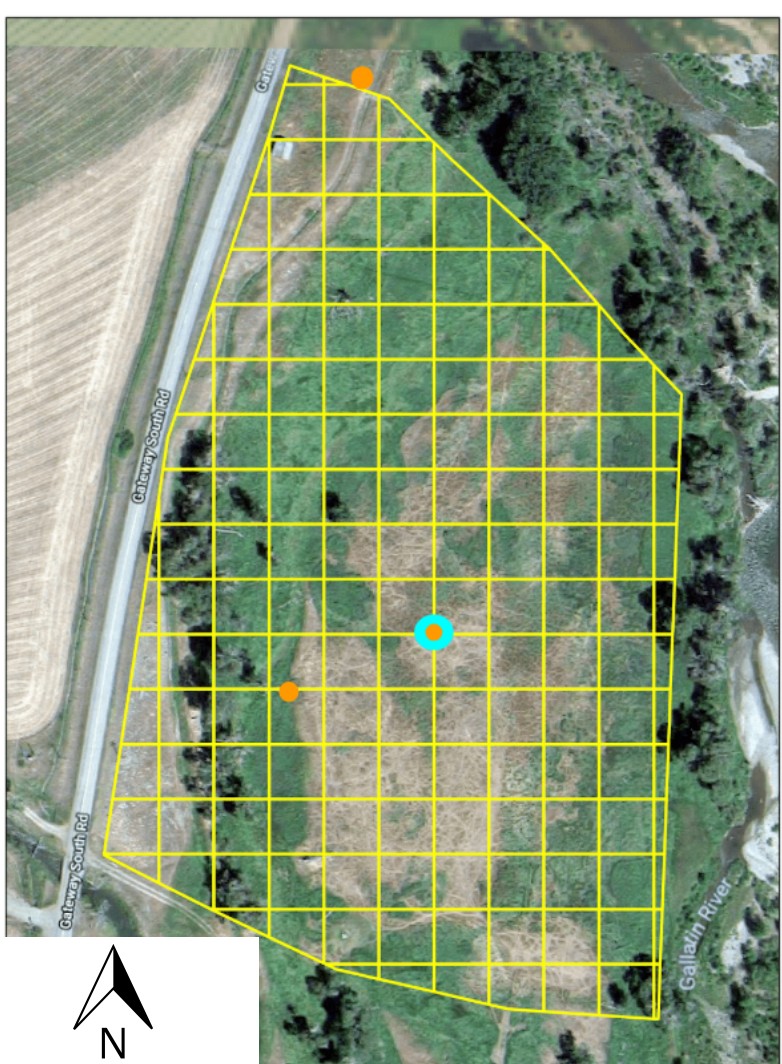


**Figure 1:** The study site near Gallatin Gateway, MT **(45.557, −111.229). Bison locations are mapped within the 20-meter grid here superimposed in yellow. The tower location is in cyan and game camera locations are indicated in orange. Background image: Google, Maxar Technologies and the USDA Farm Service Agency ©2018.**

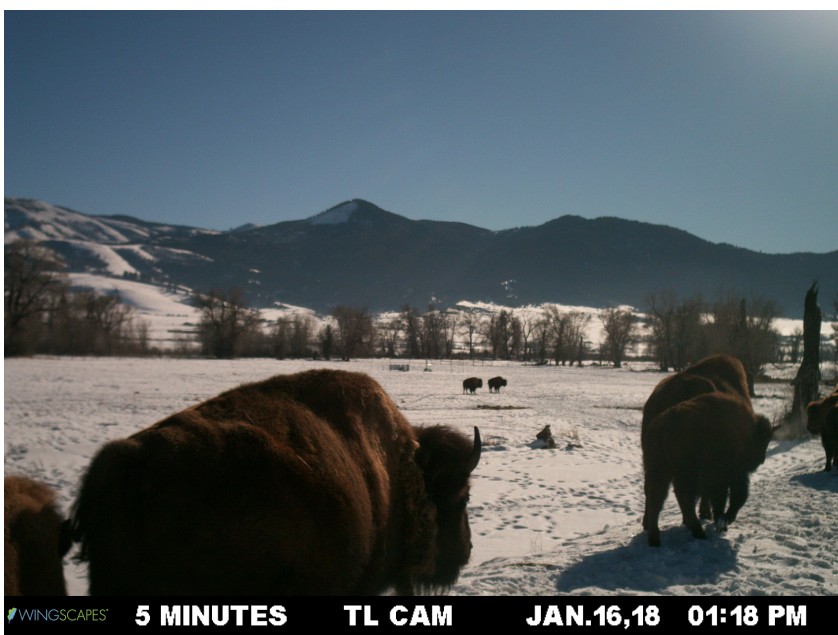

**Figure 2: A sample image of bison as viewed from the south-facing time-lapse camera located to the north of the study area (Figure 1). The eddy covariance installation is visible toward the center of the study site.**

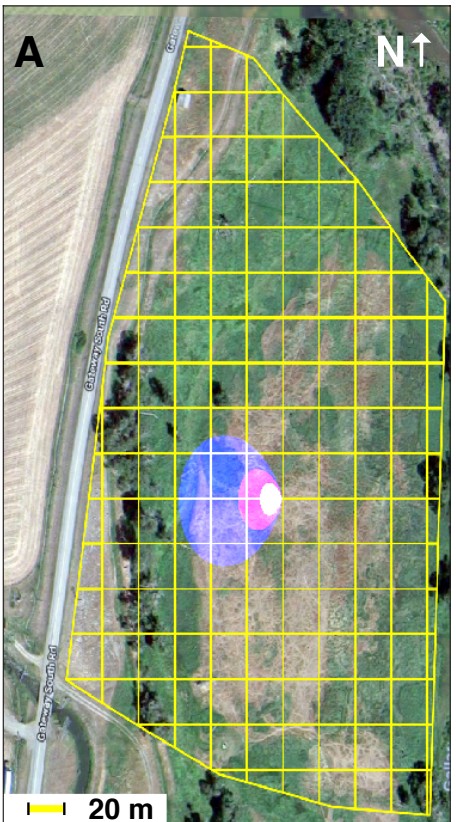 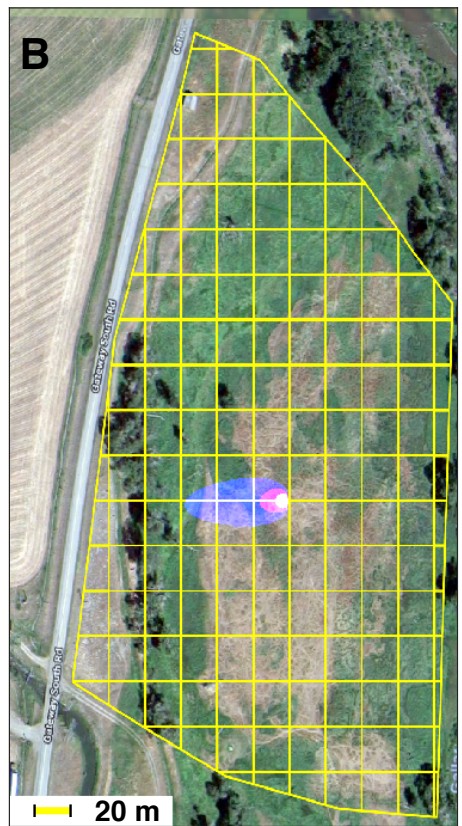

**Figure 3: An eddy covariance flux footprint calculated following (A) Hsieh et al. (2000) extrapolated to two dimensions following Detto and Katul (2006) and (B) Kljun et al. (2015) for a single 30-minute interval superimposed on the study field (Figure 1). The purple, pink, and white areas represent the 95%, 75%, and 50% footprint during 1030 AM – 1100 AM Mountain Standard Time on January 8, 2018. The fraction of the footprint in each grid box is summed for each 20-m pixel to calculate the contribution of each pixel to the total flux. Background image: Google, Maxar Technologies and the USDA Farm Service Agency ©2018.**


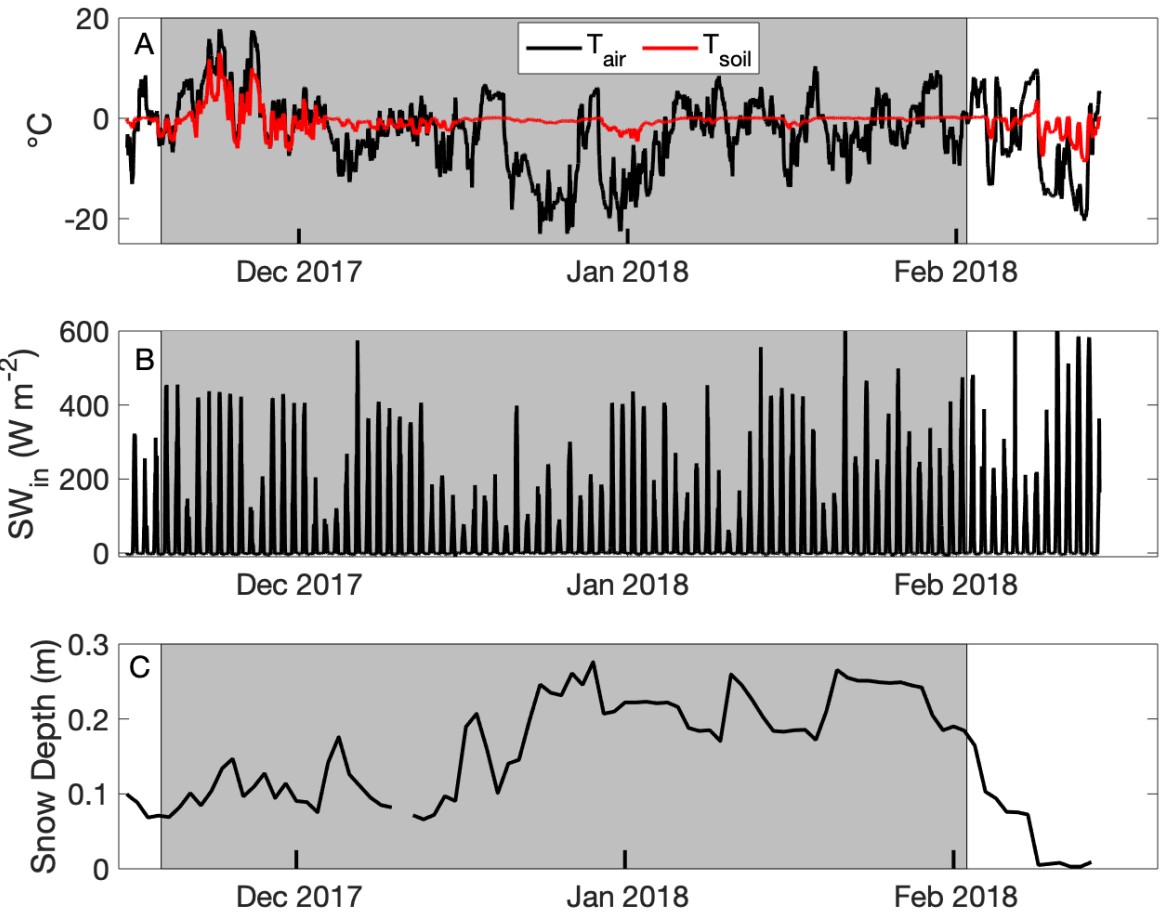


**Figure 4: (A) Air (T_air) and soil temperature (T_soil), (B) incident shortwave radiation (SW_in), and (C) snow depth from a micrometeorological tower enclosed within an electric fence on a bison pasture near Gallatin Gateway, Montana, USA. Bison were present in the pasture during the interval bounded by the grey background.**

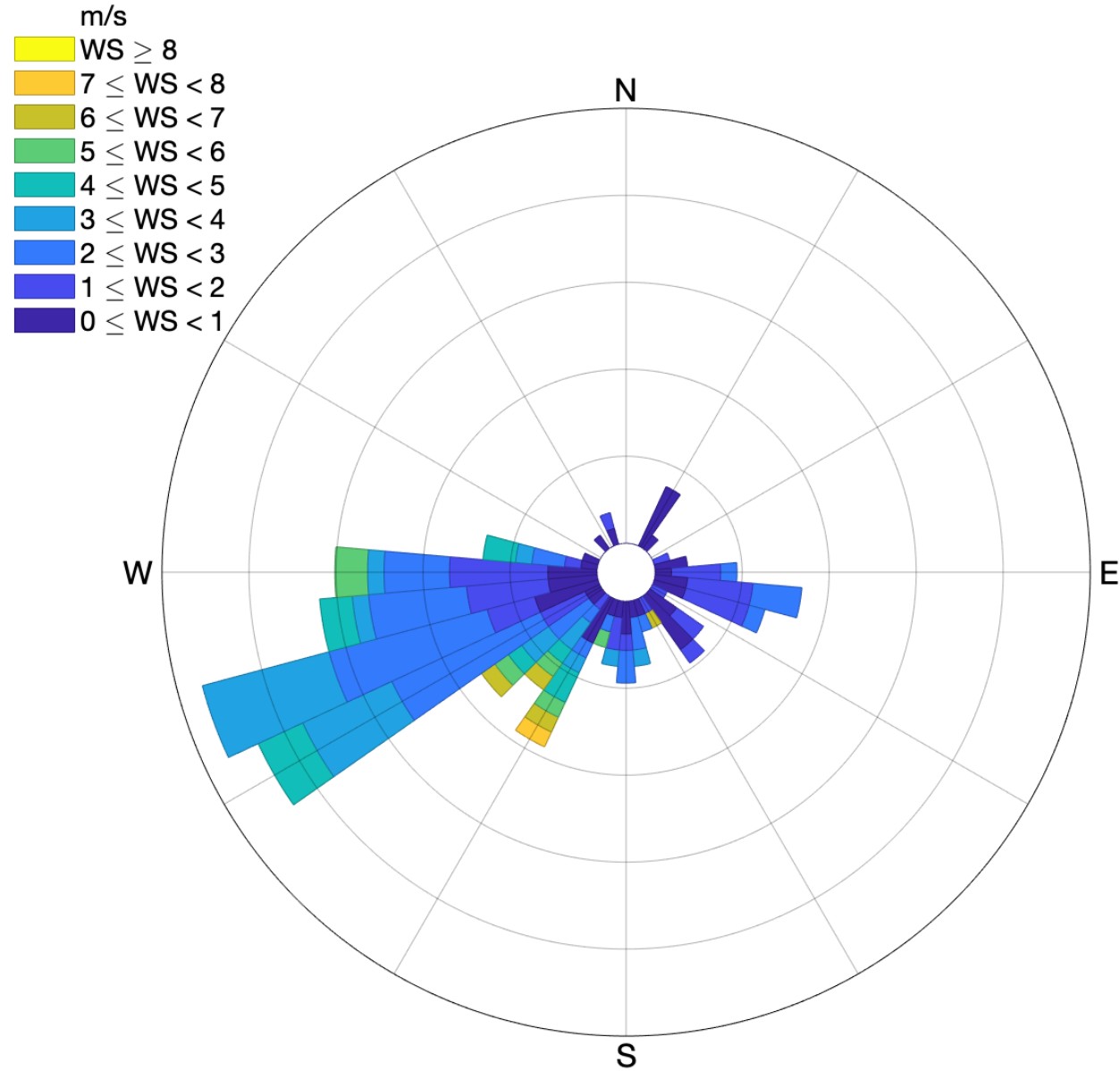

**Figure 5: A wind rose following Pereira (2020) for periods when eddy covariance measurements and bison location measurements were available. WS: wind speed.**

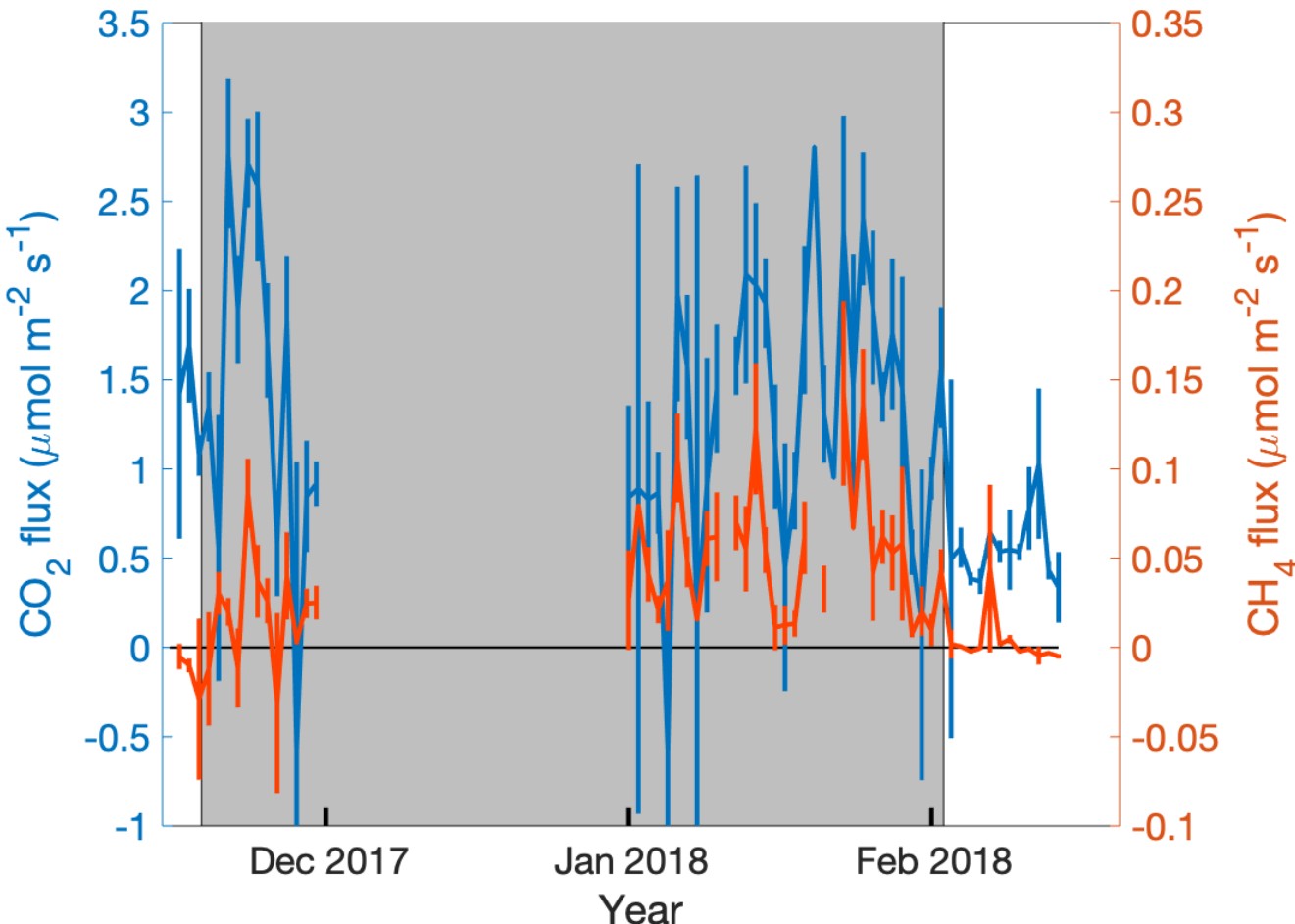

**Figure 6: The daily mean and standard error carbon dioxide and methane fluxes with standard error during daytime hours (0700-1700) from the study pasture near Gallatin Gateway, MT, USA. The gray background denotes the interval during which bison were present on the study site.**


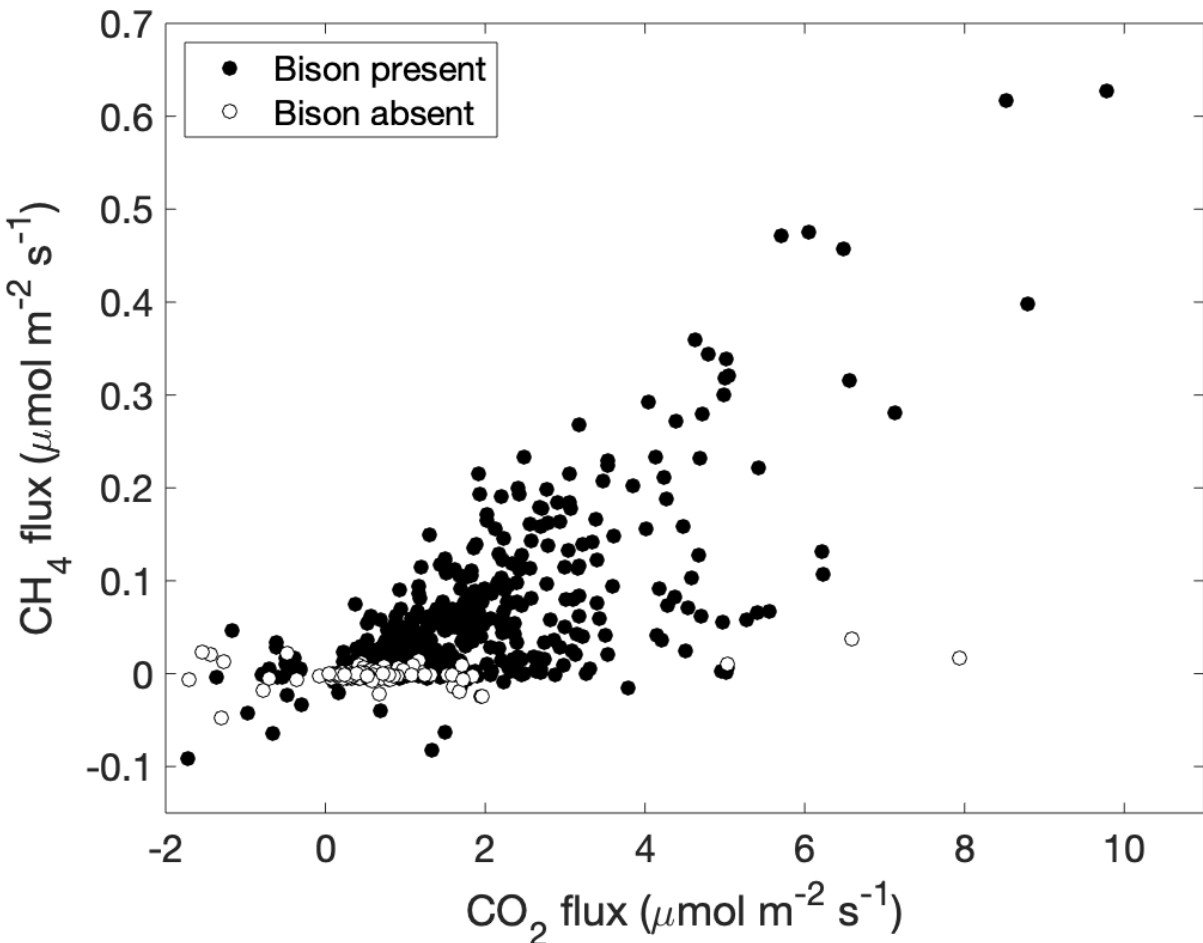

**Figure 7: The relationship between carbon dioxide and methane fluxes from the study pasture is shown for periods when bison were present (filled circles) and when bison were absent (open circles).**

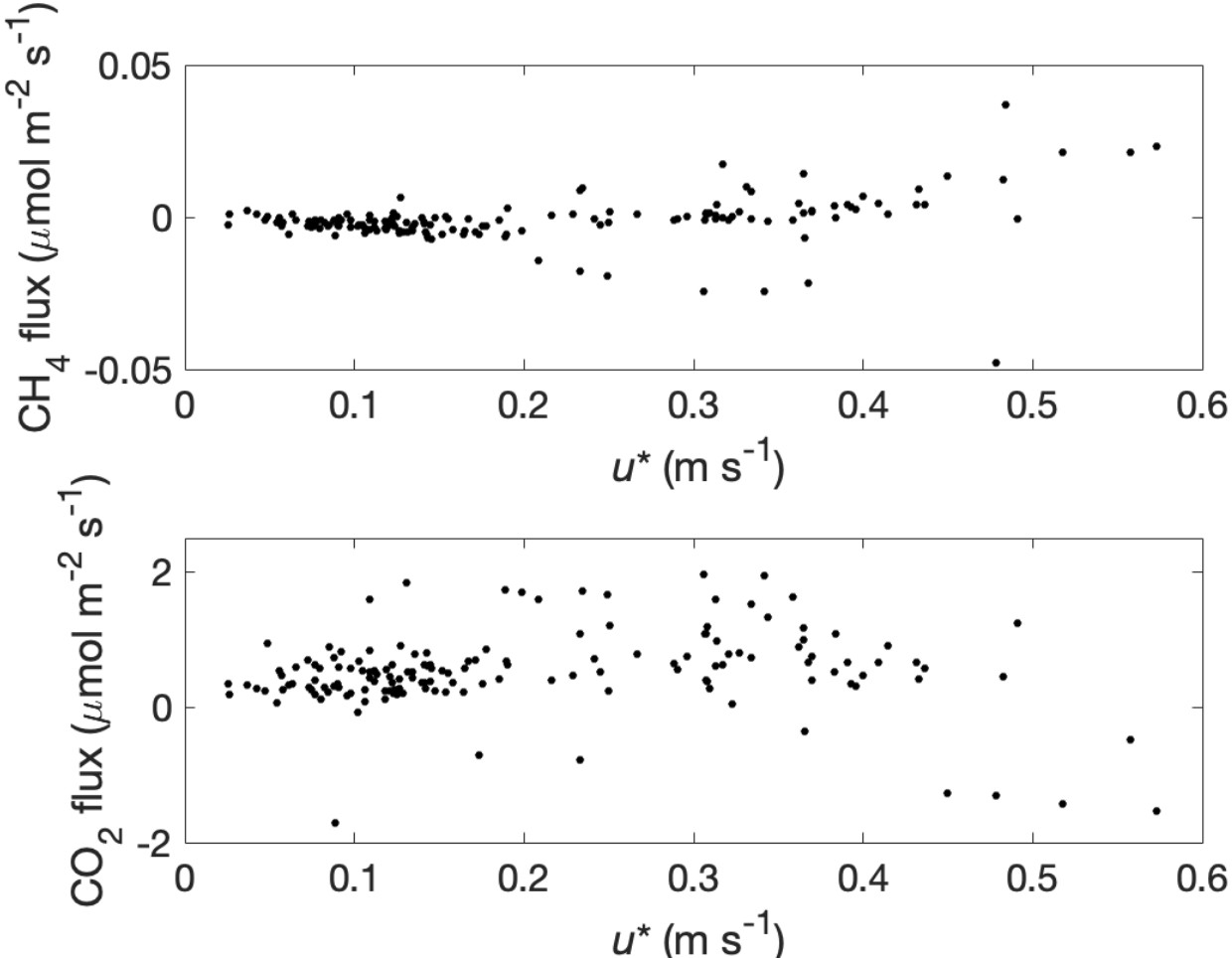

**Figure 8: Methane (A) and carbon dioxide (B) fluxes as a function of friction velocity ($u^*$) when bison were absent from the study pasture.**

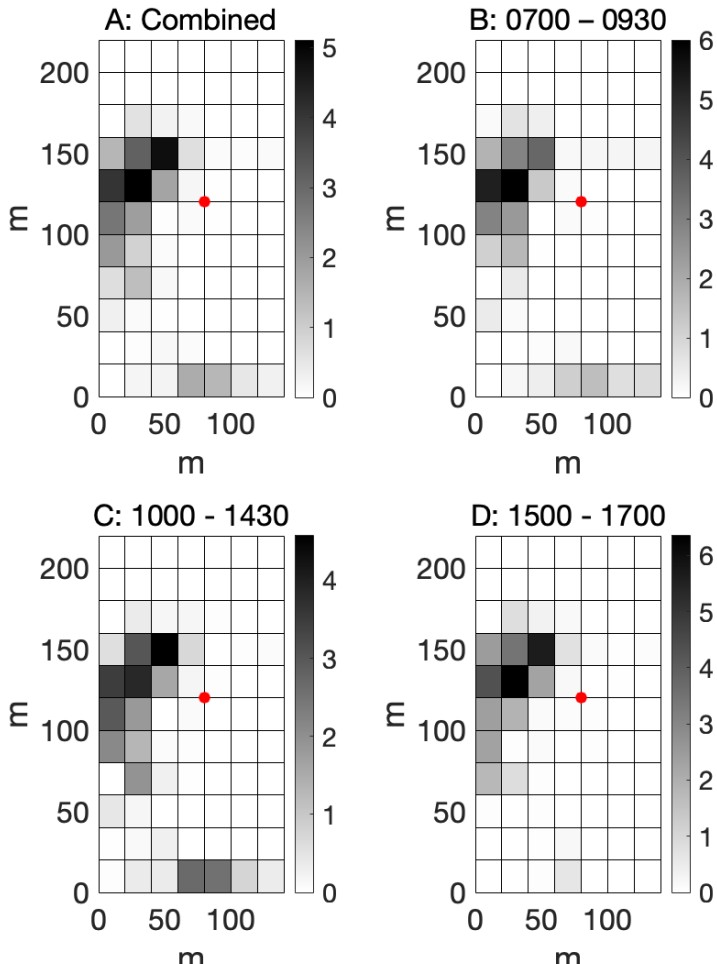

**Figure 9: Average proportional bison density for three periods of the day. Each colored pixel represents a 20-meter grid square, red dots denote the location of the eddy covariance tower, and subplot titles refer to local time. Color denotes average number of bison present in each grid cell for the 39-animal herd.**

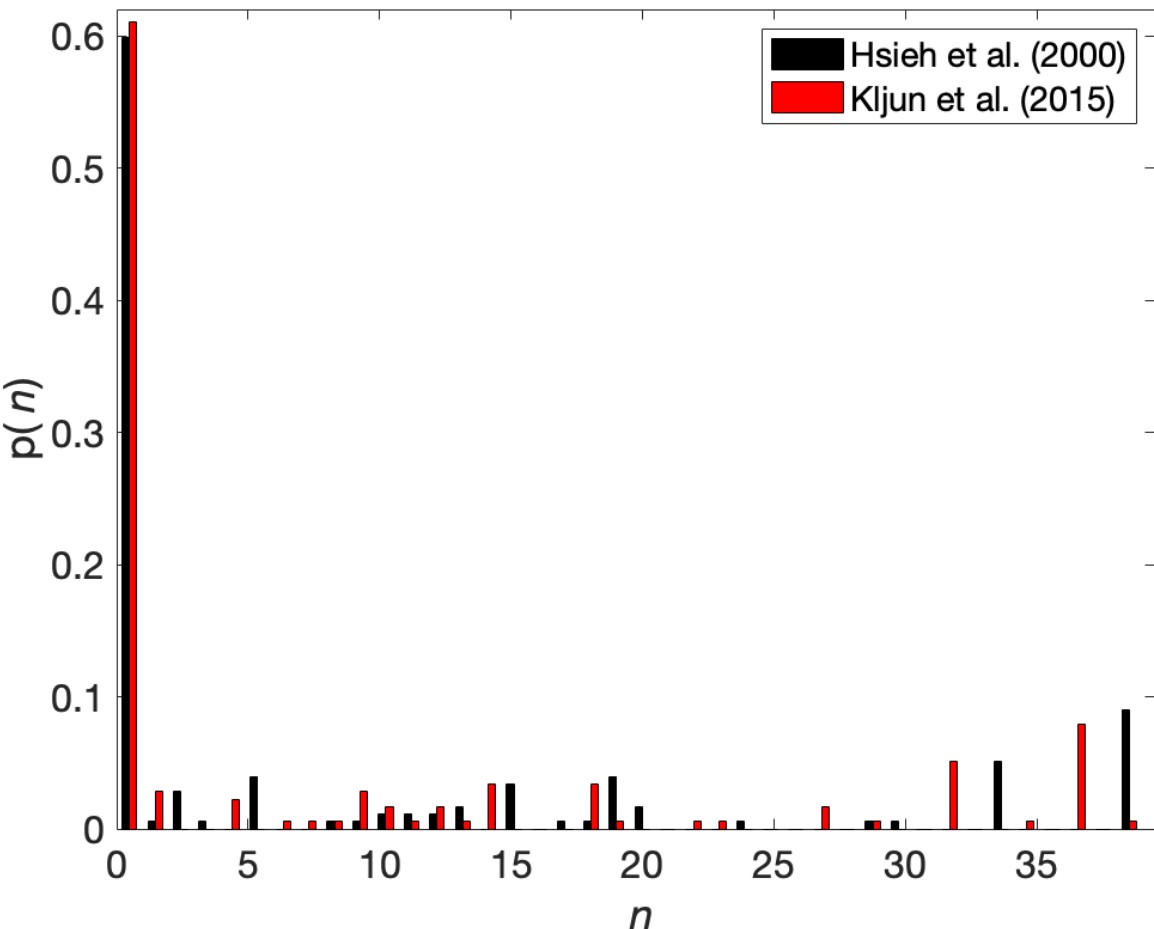

**Figure 10: The probability (p(*n*)) of the number of bison (*n*) in the 90% flux footprint for the Hsieh et al. (2000) and Kljun et al. (2015) footprint models for periods when flux measurements were made and camera imagery was available.**

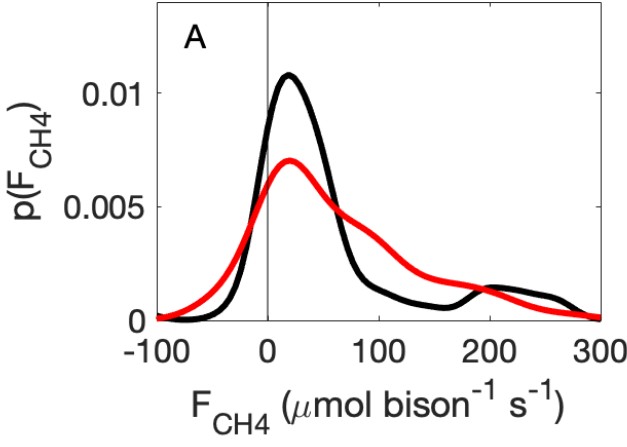

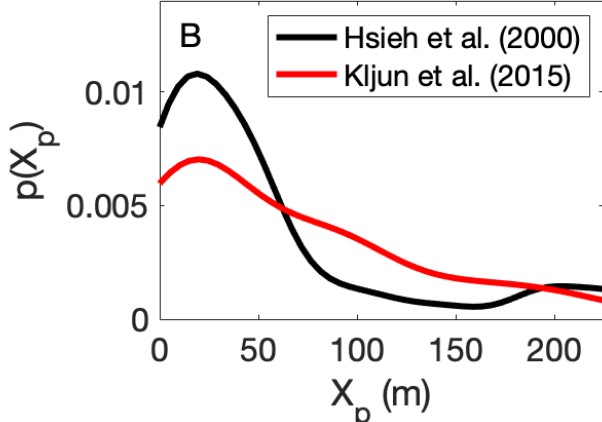


Figure 11: Kernel density estimates of the distribution (p) of (A) methane efflux ($F_{CH4}$) on a per-bison basis and (B) the peak ($X_p$) of the source-weight function for half-hourly flux footprints derived by the Hsieh et al. (2000) and Kljun et al. (2015) flux footprint models.

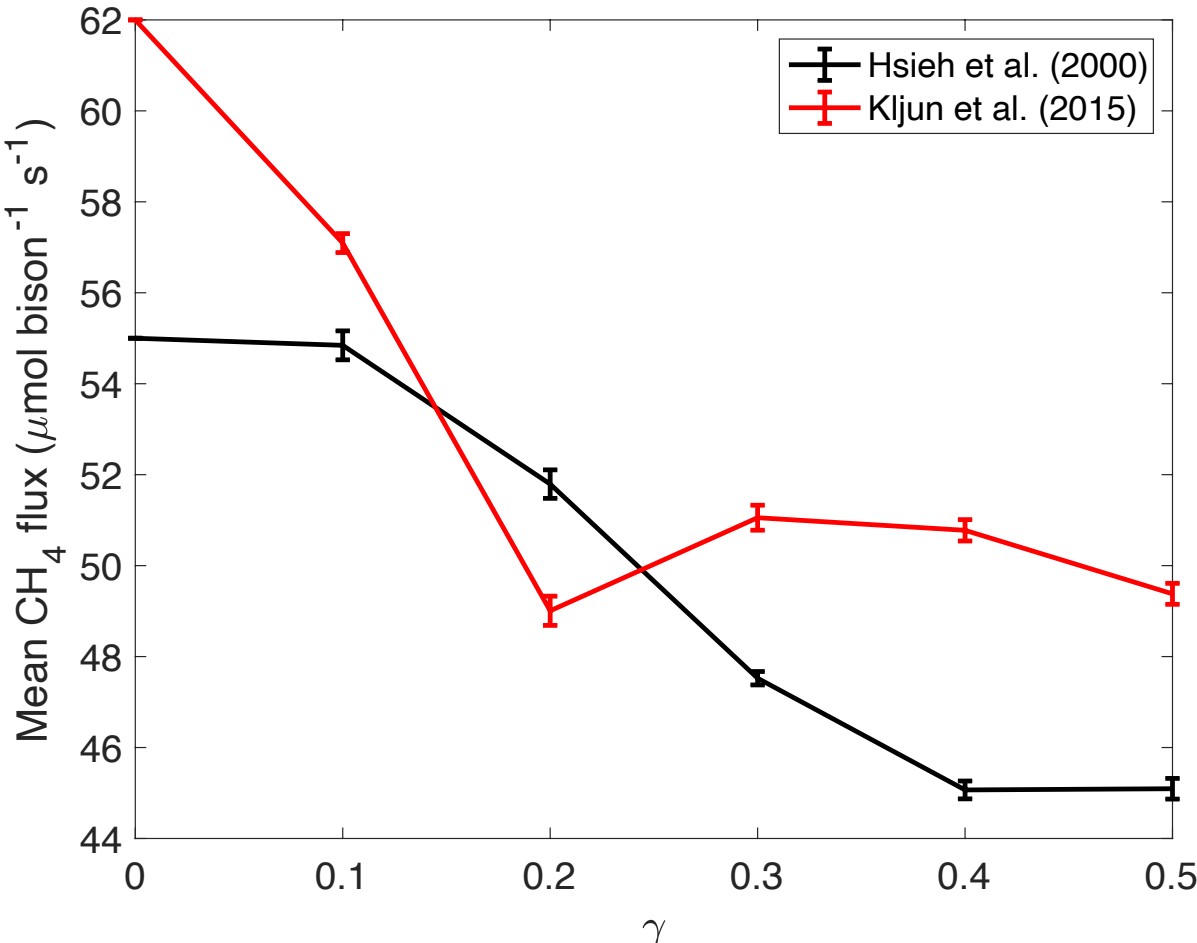

**Figure 12: The estimated mean per-bison CH$_4$ efflux from stochastic simulations of bison locations using a probability surface defined by two-dimensional Tikhonov Regularization (see Supplemental Information) for different values of the Lagrange multiplier $\gamma$. Error bars represent standard error about the mean of twenty simulations.**