# Peer review of "Methane efflux from an American bison herd"

_Biogeosciences, 2020_

## Referee Comment (RC1) · Anonymous Referee #1 · 21 Mar 2020

GENERAL COMMENTS:

In this paper, Stoy et al. estimate bison enteric emission using the eddy-covariance method, coupled with a footprint model and a cattle location method. This type of approach is under development (Felber et al., 2015; Coates et al., 2017; Dumortier et al., 2017; Prajati et al., 2018; ... all cited in the document) and has the advantages of providing an estimate in the field, integrating the animal to animal variability, having great temporal resolution and the potential to be automated. The current bison herd is small but growing and the application of such a method is especially interesting on a wild species on which the classical methods by metabolic chamber or on-animal tracers are undoubtedly more complicated to apply. Therefore the scientific interest of the paper is proven. However, this method faces technical and methodological difficulties that limit its accuracy. The choice and accuracy of the footprint model, the technical difficulty of automatic tracking of livestock location, the best way to calculate

a turbulent flux in non-stationary conditions, and the best way to determine a flux per individual based on turbulent flux and contribution to the footprint are still insufficiently investigated.

However, the paper does not present any significant advances on these points. Geolocation is carried out by manual analysis of images in the visible range, resulting in a restricted dataset of about 170 half hours, making it impossible to study a seasonal or even diurnal evolution of the emissions, a footprint model is arbitrarily chosen and is not compared to other available models and the difficulties related to non-stationarity are not addressed. The paper traces its path, in a pragmatic way admittedly, relying on choices made by other authors and not yet consolidated. An analysis of the dependence of flux on ustar in the absence of bison is proposed, with the aim of identifying a possible filtering criterion for low turbulence, but it is inconclusive in my opinion because of the low magnitude of the fluxes, both of $CH_4$ and $CO_2$. So there is little methodological input. A positive point from this point of view is the sensitivity analysis of the estimation of flux per individual to the precision of geolocation/precision of the footprint model. Some parts are however difficult to follow (e.g. smoothing of positions, see note below).

Some parts of the paper don't seem very useful to me. I am thinking in particular of the justification for the fact that the methane emissions measured do come from livestock (low background flux, i.e. from the soil/plant continuum). This is an essential part of the method, but it seems quite obvious to me for an ecosystem of this type in the winter conditions encountered. The observation of the absence of $CH_4$ flux when the bison are removed from the pasture seems to me sufficiently meaningful and I don't see the point of presenting the absence of dependence of the $CH_4$ flux on abiotic variables (radiation, temperature) to substantiate this observation. I was also hampered by some speculative passages (e.g. mechanisms of flux dependencies to u*, role of excreta, possible diurnal variability) and the perspectives are certainly well written but already known by the community.

Remains the main message that, despite the large uncertainties in the enteric emission per individual, the enteric flux is lower than that of other types of ruminants. It is stated in the introduction that since bison have a grazing behavior that favors nutrient-rich species they may have lower enteric emissions but in this study fodder is provided and is not characterized precisely, neither in terms of quantity nor in terms of quality. The reader therefore has no leads to circumstantiate this result.

I therefore feel that this article is premature and that the critical mass of original and useful information for the community is not reached at this stage. I encourage the authors to expand their dataset to allow for a statistically robust analysis of the quality of the footprint model, of the diurnal flux variability, to investigate methodological limitations in more detail and to propose explanations for low bison enteric emissions. Because I think the topic deserves a new and more robust submission when the above comments will be addressed, I also added below my specific comments, hoping it will help the authors to improve their analysis.

I would also like to point out that the shape of the paper is good, the writing is fluent, the references appropriate and the figures clear.

SPECIFIC COMMENTS:

L20: The uncertainty of 14 gCH4 day-1 bison-1 mentioned in the abstract without any additional comment is, as clearly explained in L194, only including spatial uncertainty (and I have some concerns on this point, see below) and uncertainty due to the flux summation. Information on the huge dispersion on your <f> estimates (standard deviation of 61 gCH4 day-1 bison-1 !) is not even mentioned in the abstract, which is misleading.

L28-70: Nice introduction.

L78: The composition of the herd is not specified. Age distribution could strongly influence CH4 emissions.

L118: One or two additional lines on spectral corrections would be useful. Lateral separation, reference cospectrum, magnitude of the correction factor.

L146-147: Too little information is given on the visual geolocation of bison based on the cameras. All we know is the position of the cameras and "manually attributing bison locations to squares in a 20m grid". How can you assign a distance to the mast with cameras that have no high-angle views and no distance markers in the different azimuths? Are the images from the different cameras combined to triangulate the positions of each individual? And how are the 6 positions averaged at half an hour? The authors propose an uncertainty of 20m on this estimate, which seems small in the absence of details on how to proceed.

L151-155: The paper is not self-standing on the point of "2D Tikhonov regularization". More information is needed so that the reader can understand the concept without having to read the reference assiduously. I do not master this technique but when I see that this spatial smoothing results in redistributing 3 individuals from the group at (x=40-80m,y=80-100m) to a distant group on the example of fig 4, I wonder about its relevance to simulate possible errors of location or footprint function.

L142: The approach used to determine <f> gives an estimate per half hour. However, the half hours with low contribution to the footprint will show a large dispersion, as this term is used in the denominator in eq. 3. Did the authors try to determine <f> rather by flux regression vs. contribution to the footprint?

L148-150: I don't think that shifting everything by 1 grid-square cell in each cardinal direction can simulate a systematic error of the footprint model. Proceeding in this way, the impact on the estimation of the mean of the half-hourly <f> will be smoothed. I would understand better if it was systematically shifted by 1 grid-square farther/closer with respect to the mast (modify r in polar coordinates).

L184-187: It is not clear whether Fig. 10 only shows the locations for the 173 half-hourly periods where CH4 fluxes are also available or whether it is the 444 half-hourly

periods with camera tracking. The first option seems less misleading to me.

L184-187: The forage was not brought in the direction of the prevailing winds. As a result, cattle are often on the sides of the footprint. Is Hsieh's model reliable under these conditions? You suggest that the analytical models were validated for this type of exercise (L249) but it was not the Hsieh model.

L196: You should explain how you combine the spatial uncertainty and the uncertainty due to long-term methane flux sums (but annual sums in Deventer et al., 2019, what is the logic behind using it here?)

L204: The Hogan's publication is 17 years old already. You should rely on more recent literature. Also, the 60 kgCH4 per year per animal for range cattle is an average over very contrasted cattle nature. It would be useful to be more precise.

L238-241: not convinced by the statistical reliability of this assertion. Since it seems to be the case for you too, Figure 13 should be removed.

Fig 5: Is it really necessary to show (and use in statistical analyses) both SW and Rnet?

TECHNICAL CORRECTIONS:

L191-193: the range 36-44 is repeated twice. Probably a typo?

L196: gCH4 bison-1 day-1 instead of gCH4 m-2 day-1 !!!

L211: 'negative' instead of 'positive'

L494: something is wrong in this sentence.

Fig 6: For better readability, the tower should be the origin of the spatial scale. Also in fig 10.

Fig 7: add ticks for the x scale.

Fig 9: ustar should be in m s-1

[Figure]

---

## Referee Comment (RC2) · Anonymous Referee #2 · 24 Mar 2020

Review for "Methane efflux from an American bison herd"

General comments

The manuscript "Methane efflux from an American bison herd" from Stoy et al. presents winter CH4 fluxes from a bison grazing system combined with a flux footprint analysis to estimate average CH4 fluxes per animal and day. It addresses the interesting scientific question on the magnitude of bison emissions. The data is presented in a clear structure and easy-to-follow writing style. The manuscript uses methods which have been shown with varying success elsewhere (e.g. Felber et al., 2015; Coates et al., 2017). While other authors used also automatic GPS tracking (e.g. Felber et al 2016 in AEE) the authors manually attributed the animals to a raster. Acknowledging the difficulty to assess a system of wild animals, the method used can be seen as useful first step to quantify bison emissions. A main methodological issue is that the flux uncer-

tainty is underestimated. Knowing that the different footprint models give very different results on which your approach relies upon – to better depict the uncertainty, it would be useful to analyze sensitivity of the CH4 flux per animal to different footprint models in order to include this uncertainty in the presented SE. The study lacks conclusiveness regarding the bison emission estimate: If the results were robust - What was the reason for the low CH4 emissions from bisons compared to average cattle emissions?

Specific comments

L 21 "Emission estimates are subject to spatial uncertainty in bison location measurements and the flux footprint, but from our measurements there is no evidence that bison methane emissions exceed those from cattle. We caution however that our measurements were made during winter and that evening measurements of bison distributions were not possible using our approach." The sentence does not make sense. "but" indicates a contrast, while no significant differences are exactly a result of high spatio-temporal variability/and considerable measurement uncertainty.

Please rather give the exact numbers $\pm$ SE for both estimates, for so the readers get an idea of what it means that no differences were found.

L 25 Eddy covariance is a promising technique for measuring ruminant methane emissions in conventional and alternate grazing systems and can be used to compare them going forward. RC: The sentence is not really saying much that was not known before. Rather state a concluding sentence from what you found.

Introduction

RC: L43: Add one sentence about: What is known about methane emissions from energy-dense/high-quality versus low-energy/low-quality grass for cattle?

L46: "Methane is a highly potent greenhouse gas and has about 3.7 times the global warming potential of carbon dioxide on a per- mole basis (Lashof and Ahuja, 1990)." RC: I guess you overlooked some major updates since the nineties – please cite the

most recent IPCC report (2014). The number(s) there are considerable higher...

L49: Between 30 and 40 percent of anthropogenic methane emissions are due to enteric fermentation in livestock

L60 "The important role of bison to past methane fluxes suggests that current their role in the global methane budget must be understood as their populations increase." The sentence does not make sense, improve spelling/grammar.

L 63: 30 L per kg dry food intake – how does this compare to measurements from cattle? The number is not very meaningful without comparison as a reference

Methods

If you provide hay, how are the feeding values typical for what they would eat otherwise? I would guess the hay represents rather an average, not particularly species selected to be nutrient-rich.

The methodology for deriving bison location is not clearly described. The perspective of the cameras gives a highly skewed picture. From the description the bison attribution to a grid-cell is not comprehensible. Please describe precisely what you did.

How can you justify a shift by a grid-cell of 20 m in each direction is sufficient to represent spatial inaccuracies?

Please explain to the reader the two-dimensional Tikhonov Regularization (& Lagrange multiplier) in a methods paragraph.

The methods section on the flux calculations could be more specific, i.e. state the respective thresholds and parameters used.

The paper would benefit from some numbers indicating: How many datapoints are actually available with e.g. > 20 bisons placed in the area of 60% flux contribution footprint area.

Results

It necessary to state that winter methane fluxes in the system without bisons are insignificant, as this is a basis for the whole calculation. Still, there are many words spent on this in the results and discussion, I think that this adds not much to the content of the paper.

Fig. 7: include the daily variability of fluxes

L211 negative not positive

In the highly skewed distribution (Fig 11), it is getting obvious that the SE does not represent well the uncertainties. Consider reporting quantiles of the distribution which then reflect the higher uncertainties towards higher CH4 flux values.

It would be useful and interesting to repeat the measurements with the fodder source placed in the major footprint area.

From Fig 3 and Fig 6 it becomes clear how little overlap there is between bison presence in the footprint. How would the flux estimates look like if you just choose the occasions when the joint presence of many bisons overlaps with the core (i.e. 50% flux contribution) footprint area for a certain time? Such an analysis could enhance the understanding of how robust your estimate is.

Discussion

Give an approximate estimate of the bulk uncertainties inherent to the flux calculations in the discussion section.

It remains unclear if the low CH4 fluxes for bison fluxes is a result of methodology (spatial distribution, flux footprint uncertainty, non-stationary conditions) and possibly (but probably of much less importance) also other confounding factors (fodder composition).

In the discussion, it is necessary to more specifically elaborate on why bison CH4 emissions should be that low, what can be reasons/mechanisms behind it?
The methodological issues seem to dominate the outcome of the paper and I lack of confidence in the estimated uncertainty.

---

## Author Comment (AC1) · 29 Apr 2020

GENERAL COMMENTS:
In this paper, Stoy et al. estimate bison enteric emission using the eddy-covariance method, coupled with a footprint model and a cattle location method. This type of approach is under development (Felber et al., 2015; Coates et al., 2017; Dumortier et al., 2017; Prajati et al., 2018; ... all cited in the document) and has the advantages of providing an estimate in the field, integrating the animal to animal variability, having great temporal resolution and the potential to be automated. The current bison herd is small but growing and the application of such a method is especially interesting on a wild species on which the classical methods by metabolic chamber or on-animal tracers are undoubtedly more complicated to apply. Therefore the scientific interest of the paper is proven. However, this method faces technical and methodological difficulties that limit its accuracy. The choice and accuracy of the footprint model, the technical difficulty of automatic tracking of livestock location, the best way to calculate a turbulent flux in non-stationary conditions, and the best way to determine a flux per individual based on turbulent flux and contribution to the footprint are still insufficiently investigated.

However, the paper does not present any significant advances on these points. Geo- location is carried out by manual analysis of images in the visible range, resulting in a restricted dataset of about 170 half hours, making it impossible to study a seasonal or even diurnal evolution of the emissions, a footprint model is arbitrarily chosen and is not compared to other available models and the difficulties related to non-stationarity are not addressed. The paper traces its path, in a pragmatic way admittedly, relying on choices made by other authors and not yet consolidated. An analysis of the dependence of flux on ustar in the absence of bison is proposed, with the aim of identifying a possible filtering criterion for low turbulence, but it is inconclusive in my opinion be- cause of the low magnitude of the fluxes, both of CH4 and CO2. So there is little methodological input. A positive point from this point of view is the sensitivity analysis of the estimation of flux per individual to the precision of geolocation/precision of the footprint model. Some parts are however difficult to follow (e.g. smoothing of positions, see note below). Some parts of the paper don't seem very useful to me. I am thinking in particular of the justification for the fact that the methane emissions measured do come from livestock (low background flux, i.e. from the soil/plant continuum). This is an essential part of the method, but it seems quite obvious to me for an ecosystem of this type in the winter conditions encountered. The observation of the absence of CH4 flux when the bison are removed from the pasture seems to me sufficiently meaningful and I don't see the point of presenting the absence of dependence of the CH4 flux on abiotic variables (radiation, temperature) to substantiate this observation. I was also hampered by some speculative passages (e.g. mechanisms of flux dependencies to u*, role of excreta, possible diurnal variability) and the perspectives are certainly well written but already known by the community.

Remains the main message that, despite the large uncertainties in the enteric emission per individual, the enteric flux is lower than that of other types of ruminants. It is stated in the introduction that since bison have a grazing behavior that favors nutrient-rich species they may have lower enteric emissions but in this study fodder is provided and is not characterized precisely, neither in terms of quantity nor in terms of quality. The reader therefore has no leads to circumstantiate this result.

I therefore feel that this article is premature and that the critical mass of original and useful information for the community is not reached at this stage. I encourage the authors to expand their dataset to allow for a statistically robust analysis of the quality of the footprint model, of the diurnal flux variability, to investigate methodological limitations in more detail and to propose explanations for low bison enteric emissions. Because I think the topic deserves a new and more robust submission when the above comments will be addressed, I also added below my specific comments, hoping it will help the authors to improve their analysis.
I would also like to point out that the shape of the paper is good, the writing is fluent, the references appropriate and the figures clear.

*We largely disagree with this assessment but thank the Referee for the kind notes about the writing of the manuscript. We were generously allowed to measure animals from a privately-owned herd during a select period of the calendar year and did so to the best of our abilities under the reasonable condition that disturbance to the herd be minimized, hence the automated camera approach and three-month sampling period. The method to determine average bison contribution outlined in equations 1-3 is novel and builds upon previous work demonstrating that point- or near point-sources can be captured effectively using eddy covariance (Dumortier et al., 2019; Prajapati and Santos, 2019). The diurnal evolution of the flux estimates was investigated and found not to be significant. We are puzzled that our manuscript, which to our knowledge makes the first measurements of the methane emissions of non-domesticated ruminants, was insufficiently novel. It seems like the caution with which we are interpreting our measurements – for example, exploring u\* dependencies and methane efflux in the absence of bison – are being mistaken for a lack of novelty. We note that background emissions are a major source of uncertainty of the seasonal course of methane measurements from feedlot studies of cattle (e.g. Prajapati and Santos, 2019) and felt that it was important to study this.*

*That being said, we made numerous changes to the manuscript to further improve it. We added the Kljun et al. footprint model as an independent estimate of the footprint with the generous assistance of Natascha Kljun who we added as a coauthor and also added detail about the magnitude of the corrections as noted below; thank you for suggesting that we do so. We comprehensively revised the manuscript in response to reviewer comments and feel that the revision represents a marked improvement.*

L20: The uncertainty of 14 gCH4 day-1 bison-1 mentioned in the abstract without any additional comment is, as clearly explained in L194, only including spatial uncertainty (and I have some concerns on this point, see below) and uncertainty due to the flux summation. Information on the huge dispersion on your <f> estimates (standard deviation of 61 gCH4 day-1 bison-1 !) is not even mentioned in the abstract, which is misleading.

*Uncertainty was calculated by summing the uncertainty due to spatial location and adding flux measurement uncertainty. The half-hourly uncertainty mostly averages out in the daily sum; consider for example a time series of half-hourly carbon dioxide flux data of an ecosystem during the growing season that follows the expected pattern with light. Taking the average value over the course of a day will have a large standard error but each individual measurement is accurate to within the accuracy of the flux measurement.*

L28-70: Nice introduction.

*Thank you, we wanted to describe why such measurements are necessary, especially in light of the ongoing success story of bison reintroduction in the North American Great Plains for which we owe a debt of gratitude to Tribal Nations in the US and First Nations in Canada.*

L78: The composition of the herd is not specified. Age distribution could strongly influence CH4 emissions.

*Thank you for pointing this out; we asked the landowners for the age distribution of the animals and they graciously agreed with a comprehensive table that included sex, weight, and more. We plan on adding a revised version of the table below to a new Supplemental Information section. We also added text to the discussion and a new reference noting the importance of animal age (and especially size) on per-animal methane efflux. Information from the landowner also clarified a question that we had about the number of animals in the pasture. Staff had originally told us that there were 40 animals but records indicate 39, which aligns better with the numbers from counts. We adjusted our location maps accordingly and re-ran the analyses.*

| SEX | BIRTH YEAR | WEIGHT (lbs.) | Weight (kg) | WEIGHT DATE | PREGNANCY STATUS |
|-----|-----------|---------------|-------------|-------------|------------------|
| F | 2010 | 1030 | 467 | 11/16/17 | Y |
| F | 2010 | 924 | 419 | 11/16/17 | Y |
| F | 2010 | 944 | 428 | 11/16/17 | Y |
| F | 2010 | 1055 | 479 | 11/16/17 | Y |
| F | 2010 | 1125 | 510 | 11/16/17 | Y |
| F | 2010 | 1050 | 476 | 11/16/17 | Y |
| F | 2010 | 1085 | 492 | 11/16/17 | Y |
| F | 2010 | 1000 | 454 | 11/16/17 | Y |
| F | 2010 | 1250 | 567 | 11/16/17 | Y |
| F | 2010 | 1050 | 476 | 11/16/17 | Y |
| F | 2010 | 1095 | 497 | 11/16/17 | Y |
| F | 2010 | 1015 | 460 | 11/16/17 | Y |
| F | 2010 | 976 | 443 | 11/16/17 | Y |
| F | 2010 | 958 | 435 | 11/16/17 | Y |
| F | 2010 | 940 | 426 | 11/16/17 | Y |
| F | 2010 | 1050 | 476 | 11/16/17 | Y |
| F | 2010 | 906 | 411 | 11/16/17 | Y |
| M | 2012 | 1425 | 646 | 11/16/17 | |
| M | 2012 | 1545 | 701 | 11/16/17 | |
| F | 2014 | 840 | 381 | 11/16/17 | Y |
| F | 2014 | 904 | 410 | 11/16/17 | Y |

| F | 2016 | 736 | 334 | 11/16/17 | |
| F | 2017 | 242 | 110 | 11/16/17 | |
| F | 2017 | 318 | 144 | 11/16/17 | |
| M | 2017 | 353 | 160 | 11/16/17 | |
| F | 2017 | 367 | 166 | 11/16/17 | |
| M | 2017 | 305 | 138 | 11/16/17 | |
| M | 2017 | 335 | 152 | 11/16/17 | |
| M | 2017 | 325 | 147 | 11/16/17 | |
| M | 2017 | 403 | 183 | 11/16/17 | |
| F | 2017 | 212 | 96 | 11/16/17 | |
| M | 2017 | 458 | 208 | 11/16/17 | |
| M | 2017 | 230 | 104 | 11/16/17 | |
| M | 2017 | 360 | 163 | 11/16/17 | |
| F | 2017 | 279 | 127 | 11/16/17 | |
| M | 2017 | 299 | 136 | 11/16/17 | |
| M | 2017 | 364 | 165 | 11/16/17 | |
| M | 2017 | 278 | 126 | 11/16/17 | |
| F | 2017 | 279 | 127 | 11/16/17 | |

L118: One or two additional lines on spectral corrections would be useful. Lateral separation, reference cospectrum, magnitude of the correction factor.

*We added information about the magnitude of the correction factors to $CH_4$ fluxes (on the order of 14% when bison were present) and now note the LI-7700 which was offset in the horizontal by 22 cm (which is less than the dimensions of the optical path of the instrument at 50 cm) and 0 cm in the vertical.*

L146-147: Too little information is given on the visual geolocation of bison based on the cameras. All we know is the position of the cameras and "manually attributing bison locations to squares in a 20m grid". How can you assign a distance to the mast with cameras that have no high-angle views and no distance markers in the different azimuths? Are the images from the different cameras combined to triangulate the positions of each individual? And how are the 6 positions averaged at half an hour? The authors propose an uncertainty of 20m on this estimate, which seems small in the absence of details on how to proceed.

*Features in the field made it reasonably easy to determine locations (with uncertainty) and we took averages of the six positions for the half-hourly location estimates. Because these and all measurements have uncertainty we decided that it would be appropriate to perform the sensitivity analyses on locations to ensure that our uncertainty is not underestimated. We are more interested here in correctly characterizing uncertainty than pretending that our eddy covariance measurements of non-domesticated ruminants in wintertime in Montana have low uncertainty.*

*The bison location work has an interesting caveat. It makes little difference to the flux calculation. Take for example an extremely conservative viewpoint of bison location that assumes almost no ability to attribute bison to a particular location outside of a general location on one side of the tower. Such a situation is demonstrated on the right-hand side of the figure below where the Lagrange multiplier is set to 20 for a bison location estimate that happens to be from Jan. 22 at 11 am when bison were clustered in a clump, shown in the subplot below on the left, which is pretty easy to observe (see for example Figure 2 in the manuscript).*

[Figure]

*If we extend the Tikhonov Regularization analysis to a Lagrange multiplier of 20, representing a very crude visual guess as to the bison location as demonstrated above, the average per-bison methane flux value over the measurement period is about 30 μmol $CH_4$ bison$^{-1}$ s$^{-1}$ as demonstrated below. This is admittedly much smaller than the derived estimate of about 38 μmol $CH_4$ bison$^{-1}$ s$^{-1}$, but we also feel that we can place bison on the landscape using 8 cameras much more accurately than such a wild guess. That being said, the location attribution approach results in uncertainty, and our sensitivity analyses is designed to characterize this uncertainty. Despite this we engaged in the independent footprint estimates as suggested. Initial results are promising and helped us further characterize the uncertainty in our observations.*

[Figure]

*We undertook the rather comprehensive spatial uncertainty estimates because we were fully aware that the measurements had uncertainty and we sought to be exceedingly honest about the uncertainty in per-animal flux measurements that resulted. Such honesty should not be interpreted as lack of rigor.*

L151-155: The paper is not self-standing on the point of "2D Tikhonov regularization". More information is needed so that the reader can understand the concept without having to read the reference assiduously. I do not master this technique but when I see that this spatial smoothing results in redistributing 3 individuals from the group at (x=40-80m,y=80-100m) to a distant group on the example of fig 4, I wonder about its relevance to simulate possible errors of location or footprint function.

*We revised section 2.6 to further describe Tikhonov regularization approach used to interpret the bison location estimate with caution. To be honest we were delighted that regularization had the effect of redistributing individuals to different groups (this is entirely due to rounding to full integers), which we felt shared similarities to the tendency of animals to move between different groups as part of their social behavior.*

L142: The approach used to determine <f> gives an estimate per half hour. However, the half hours with low contribution to the footprint will show a large dispersion, as this term is used in

the denominator in eq. 3. Did the authors try to determine <f> rather by flux regression vs. contribution to the footprint?

*We did not determine <f> by flux regression in the manuscript as we believe that this would not fully incorporate the dynamic that exists between bison locations and the flux footprint. (More directly, we feel that it is incorrect to do so and are surprised at this suggestion.) An earlier effort to estimate $CH_4$ flux as a function of bison count estimated the effective number of bison in the footprint but the regression was poorly constrained and subsequent work improved the flux footprint location. Instead – and admittedly we should have been clear about this in the original manuscript – we thresholded the dataset to exclude per-animal flux values using outlier identification which we subsequently revised now describe in more detail in the revised manuscript.*

L148-150: I don't think that shifting everything by 1 grid-square cell in each cardinal direction can simulate a systematic error of the footprint model. Proceeding in this way, the impact on the estimation of the mean of the half-hourly <f> will be smoothed. I would understand better if it was systematically shifted by 1 grid-square farther/closer with respect to the mast (modify r in polar coordinates).

*We somewhat disagree, because in our case this is equivalent to shifting the bison distribution in the opposite direction. But we do agree that the inference is reversed in this case and that the flux footprint location likely has less spatial bias than the bison count estimation. We re-analyzed the data by shifting the bison location estimates rather than the flux footprint location estimates; thank you for the suggestion.*

L184-187: It is not clear whether Fig. 10 only shows the locations for the 173 half- hourly periods where CH4 fluxes are also available or whether it is the 444 half-hourly periods with camera tracking. The first option seems less misleading to me.

*We did not intend to be misleading; rather we wanted to demonstrate the diurnal behavior of the bison to demonstrate to the reader that they usually congregate in directions upwind of the tower. We re-created the figure to only include observations for which fluxes were measured.*

L184-187: The forage was not brought in the direction of the prevailing winds. As a result, cattle are often on the sides of the footprint. Is Hsieh's model reliable under these conditions? You suggest that the analytical models were validated for this type of exercise (L249) but it was not the Hsieh model.

*The forage was delivered by employees of the private ranch due west of the tower in a place where the field was accessible from the road. Wind most commonly arrived from the southwest, but there is a secondary peak of wind directions from due west during periods when methane measurements were made (see below). Bison tended to congregate to the south, southwest, and west of the tower such that there was considerable overlap between bison and the flux footprint. We were fortunate in this regard because we placed the tower in the center of the field (Figures 1 and 3) as we felt that it was the best practice for flux measurements and bison tended to congregate in the dominant wind directions, noting that the footprint can be rather broad due to*

*the variance of the lateral wind velocity as demonstrated in Figure 3. To further extend the sensitivity analysis on the flux calculations, we added the Kljun et al. model to the analysis as an independent and additional assessment of the results.*

[Figure]

L196: You should explain how you combine the spatial uncertainty and the uncertainty due to long-term methane flux sums (but annual sums in Deventer et al., 2019, what is the logic behind using it here?)

*We combined uncertainty values by summing variances then computing the standard deviation.*

L204: The Hogan's publication is 17 years old already. You should rely on more recent literature. Also, the 60 kgCH4 per year per animal for range cattle is an average over very contrasted cattle nature. It would be useful to be more precise.

*Older values are not necessarily less reliable but we added newer references including Prajapati and Santos, 2019, which was published as we were preparing the manuscript and escaped our initial notice, thank you for the suggestion.*

L238-241: not convinced by the statistical reliability of this assertion. Since it seems to be the case for you too, Figure 13 should be removed.

*We spent quite a bit of time trying to interpret if methane efflux differed over the course of the day as a function of their preferred feeding times but results were not conclusive. We removed Fig. 13 and now simply note in the text that significant diurnal methane flux patterns were not observed.*

Fig 5: Is it really necessary to show (and use in statistical analyses) both SW and Rnet?

*From the observations these variables differ rather strongly due in large part to the brightness of the snow and the differences between the snow surface temperature and sky temperature in the longwave. That being said, we do not use net radiation in subsequent analyses and removed the subplot to make the figure less busy.*

TECHNICAL CORRECTIONS:
L191-193: the range 36-44 is repeated twice. Probably a typo?

*Thank you for pointing this out. We looked into it and it happens that both approaches independently arrived at the same range. We assume that this is due to chance.*

L196: gCH4 bison-1 day-1 instead of gCH4 m-2 day-1 !!!

*This is correct, thank you for pointing out this error.*

L211: 'negative' instead of 'positive'

*Referee #2 also noted this error and it is now corrected, thank you for the careful read.*

L494: something is wrong in this sentence.

*The sentence was unnecessarily wordy. We re-wrote it to state 'Figure 3: An eddy covariance flux footprint calculated following Hsieh et al. (2000) and Detto and Katul (2006) at 1 m resolution for a single 30-minute interval superimposed on the study field (Figure 1).'*

Fig 6: For better readability, the tower should be the origin of the spatial scale. Also in fig 10.

*This is an interesting point and we carefully considered it but decided to keep the figure as is because it aligns with the grid in Figures 1 and 3 that we used to attribute bison locations. We did change the font size of the figure to have more information along the x-axis.*

Fig 7: add ticks for the x scale.

*We agree that tick marks on the x axis are an improvement and added these along with standard error bars as recommended by Referee #2.*

Fig 9: ustar should be in m s-1
*Our apologies, this is clearly a typo on our part. The figure has been revised.*

---

## Author Comment (AC2) · 29 Apr 2020

Review for "Methane efflux from an American bison herd"
General comments
The manuscript "Methane efflux from an American bison herd" from Stoy et al. presents winter CH4 fluxes from a bison grazing system combined with a flux footprint analysis to estimate average CH4 fluxes per animal and day. It addresses the interesting scientific question on the magnitude of bison emissions. The data is presented in a clear structure and easy-to-follow writing style. The manuscript uses methods which have been shown with varying success elsewhere (e.g. Felber et al., 2015; Coates et al., 2017). While other authors used also automatic GPS tracking (e.g. Felber et al 2016 in AEE) the authors manually attributed the animals to a raster. Acknowledging the difficulty to assess a system of wild animals, the method used can be seen as useful first step to quantify bison emissions. A main methodological issue is that the flux uncertainty is underestimated. Knowing that the different footprint models give very different results on which your approach relies upon – to better depict the uncertainty, it would be useful to analyze sensitivity of the CH4 flux per animal to different footprint models in order to include this uncertainty in the presented SE. The study lacks conclusiveness regarding the bison emission estimate: If the results were robust - What was the reason for the low CH4 emissions from bisons compared to average cattle emissions?

*Thank you for the insightful comments, addressing them improved the manuscript. We added the Kljun et al. footprint model to further characterize uncertainty as suggested. A side-by-side (or similar) comparison between cattle and bison systems would be necessary to understand the mechanisms causing any discrepancy, and we hope that the present manuscript helps justify such an extensive undertaking.*

Specific comments
L 21 "Emission estimates are subject to spatial uncertainty in bison location measurements and the flux footprint, but from our measurements there is no evidence that bison methane emissions exceed those from cattle. We caution however that our measurements were made during winter and that evening measurements of bison distributions were not possible using our approach." The sentence does not make sense. "but" indicates a contrast, while no significant differences are exactly a result of high spatio- temporal variability/and considerable measurement uncertainty. Please rather give the exact numbers ± SE for both estimates, for so the readers get an idea of what it means that no differences were found.

*We removed the passage for clarity because no direct measurements of cattle were made. Finding adjacent or proximal bison and cattle grazing systems to measure has been an ongoing challenge, and one that we hope to address in future research.*

L 25 Eddy covariance is a promising technique for measuring ruminant methane emissions in conventional and alternate grazing systems and can be used to compare them going forward. RC: The sentence is not really saying much that was not known before. Rather state a concluding sentence from what you found.

*We feel that the passage as written is accurate because eddy covariance has not been used to measure methane efflux from non-domesticated animals before. Because our study is in part a proof-of-concept that is important to demonstrate feasibility for future research efforts on non-*

*domesticated ungulates, we made the last line of the abstract more directed and now write, 'Our observations point to the need for direct comparisons of methane emissions from conventional and alternate grazing systems using eddy covariance and demonstrate the potential for using eddy covariance to measure methane efflux from non-domesticated animals.'.*

Introduction
RC: L43: Add one sentence about: What is known about methane emissions from energy-dense/high-quality versus low-energy/low-quality grass for cattle?

*We added 'and feed quality (Hammond et al., 2016)', thank you for the suggestion.*

L46: "Methane is a highly potent greenhouse gas and has about 3.7 times the global warming potential of carbon dioxide on a per- mole basis (Lashof and Ahuja, 1990)." RC: I guess you overlooked some major updates since the nineties – please cite the most recent IPCC report (2014). The number(s) there are considerable higher...

*We deleted the passage because these comparisons rely on a subjective time window and the readership of Biogeosciences is familiar with the importance of methane as a greenhouse gas.*

L49: Between 30 and 40 percent of anthropogenic methane emissions are due to enteric fermentation in livestock

*We clarified this passage to state 'current' anthropogenic methane emissions.*

L60 "The important role of bison to past methane fluxes suggests that current their role in the global methane budget must be understood as their populations increase." The sentence does not make sense, improve spelling/grammar.

*Thank you for the suggestion the passage was re-worded for clarity.*

L 63: 30 L per kg dry food intake – how does this compare to measurements from cattle? The number is not very meaningful without comparison as a reference

*These are simply the results presented by Galbraith et al. who did not measure cattle in their study. In the revised manuscript we cite cattle values from Hammond et al., 2016 who used a similar technique and found nearly identical values 16.5 g methane / kg dry matter intake (= 29.8 L methane / kg dry matter intake) when feeding dairy cattle a high maize silage; thank you for suggesting that we dig into this topic further.*

Methods
If you provide hay, how are the feeding values typical for what they would eat otherwise? I would guess the hay represents rather an average, not particularly species selected to be nutrient-rich.

*The landowners provide supplemental hay from a nearby field. Upon further request, we were sent extensive tables of the hay nutrient content and feed and we will summarize these as*

*supplemental material noting that bison were also free to graze within the pasture. In the revision we will provide a version of the following nutrition information and feeding tables. We assume that the forage is of similar quality to whatever pasture grasses the bison are eating, but we were unable to confirm this independently and the bison fed rather vigorously on the supplemental hay.*

Table 1: Composition of the first cut and second cut hay provided as a supplement to the study bison herd.

| Variable (% unless otherwise noted) | First cut | Second cut |
|---|---|---|
| Crude Protein | 9.7 | 17.2 |
| Acid detergent fiber | 47.9 | 38.3 |
| Total digestible nutrients | 48.9 | 59.7 |
| Calcium | 0.8 | 1.51 |
| Phosphorus | 0.2 | 0.21 |
| Magnesium | 0.21 | 0.32 |
| Potassium | 1.92 | 2.06 |
| Sulfur | 0.15 | 0.32 |
| Sodium | <0.011 | 0.028 |
| Zinc (mg/kg) | 14 | 15 |
| Iron (mg/kg) | 66 | 61 |
| Manganese (mg/kg) | 60 | 56 |
| Copper (mg/kg) | 7 | 9 |

Table 2: The schedule of bail delivery of first cut and second cut hay to the bison pasture.

| | First cut (number of bails) | Second cut (number of bails) |
|---|---|---|
| 1-Nov | 2 | 2 |
| 3-Nov | 2 | 2 |
| 5-Nov | 1 | 1 |
| 6-Nov | 2 | 2 |
| 8-Nov | 2 | 1 |
| 10-Nov | 2 | 1 |
| 12-Nov | 1 | 1 |
| 13-Nov | 1 | 1 |
| 14-Nov | 1 | 1 |
| 15-Nov | | 2 |
| 16-Nov | 1 | 1 |
| 17-Nov | 2 | 2 |

| | | |
|---|---|---|
| 20-Nov | | 2 |
| 22-Nov | 1 | 2 |
| 25-Nov | | 2 |
| 27-Nov | 2 | 2 |
| 29-Nov | | 2 |
| 1-Dec | | 2 |
| 3-Dec | | |
| 5-Dec | 2 | 2 |
| 8-Dec | 2 | 2 |
| 12-Dec | | 2 |
| 15-Dec | 2 | 2 |
| 19-Dec | 2 | 2 |
| 21-Dec | 2 | 2 |
| 26-Dec | 2 | 2 |
| 28-Dec | 2 | 2 |
| 31-Dec | 2 | 2 |
| 2-Jan | 2 | 2 |
| 5-Jan | 2 | 2 |
| 8-Jan | 2 | 2 |
| 11-Jan | 2 | 2 |
| 15-Jan | 2 | 2 |
| 18-Jan | 2 | 2 |
| 22-Jan | 2 | 2 |
| 26-Jan | | 2 |
| 27-Jan | 2 | |
| 29-Jan | 1 | 1 |
| 31-Jan | 1 | 1 |
| 3-Feb | 2 | 2 |
| 6-Feb | 2 | 2 |
| 10-Feb | 2 | 2 |
| 14-Feb | 2 | 2 |
| 19-Feb | 2 | 2 |
| 22-Feb | 2 | 2 |
| 26-Feb | 2 | 2 |

The methodology for deriving bison location is not clearly described. The perspective of the cameras gives a highly skewed picture. From the description the bison attribution to a grid-cell is not comprehensible. Please describe precisely what you did.

*This was a labor-intensive process. We interpreted each five-minute period from multiple cameras and used visual cues in the field (like trees in the background) to note the locations of the animals. Fortunately they were usually congregated in a group which made them relatively easy to place, but wanted to be extremely conservative with our location estimates and their impact on per-animal fluxes, hence the sensitivity analyses.*

How can you justify a shift by a grid-cell of 20 m in each direction is sufficient to represent spatial inaccuracies?

*Per the response to Referee 1, we were somewhat surprised to find that the precise spatial representation did not make a large difference in per-animal flux estimates. Roughly associating bison to general areas around the tower (following the figure below for one particular half-hour) decreased the per-animal flux estimate by only about 25%. We could be even more conservative with our uncertainty analyses but feel that the Tikhonov Regularization analysis accounts for spatial uncertainties and also provides realistic bounds on per-animal flux values that could be generated. We decided to shift the maps of bison location as an additional check on the sensitivity of the flux values to bison location to provide an even more conservative estimation of uncertainties. We feel that the resulting flux values honestly represent the inherent uncertainties in our analysis.*

Please explain to the reader the two-dimensional Tikhonov Regularization (& Lagrange multiplier) in a methods paragraph.

*We describe Tikhonov Regularization in more detail in the revised manuscript by expanding section 2.6.*

The methods section on the flux calculations could be more specific, i.e. state the respective thresholds and parameters used.

*We feel that we were reasonably clear about the flux calculations having indicated spike thresholds but we agree that we could have been more clear about necessary filtering post-processing. We revisited the logical thresholds that we applied to the original dataset after applying the Kljun et al. (2015) flux footprint model and increased the upper limit to 300 micromoles $CH_4$ $m^{-2}$ $bison^{-1}$. Doing so made a small change to average flux values that we feel more confident in because of very intermittent data and large gaps in the histogram at values greater than 300 micromoles $CH_4$ $m^{-2}$ $bison^{-1}$.*

The paper would benefit from some numbers indicating: How many datapoints are actually available with e.g. > 20 bisons placed in the area of 60% flux contribution footprint area.

*This is an interesting question but we did not feel that it would lead to clarity as each pixel in which bison are located represents a small contribution to the integrated footprint area and the per-bison methane contribution that it represents is embodied in the calculation in equations 1-3.*

Results

It necessary to state that winter methane fluxes in the system without bisons are insignificant, as this is a basis for the whole calculation. Still, there are many words spent on this in the results and discussion, I think that this adds not much to the content of the paper.

*We agree and took care to minimize the discussion of methane efflux in the absence of bison, but also felt that it was important to describe given potential methane sources in a field that is frequented by wild ungulates (who can jump the fence) and the nearby river (that is not in the dominant wind direction. We still wanted to be very diligent in noting that the field otherwise is near-neutral with respect to methane efflux. We are not sure why that the cautious approach that we take throughout the manuscript is deemed superfluous.*

Fig. 7: include the daily variability of fluxes

*Previous versions of the text included error bars that made the trends difficult to distinguish and we presented the median rather than the mean to emphasize the bulk of the trends. We worked to create a version that includes error bars and that is hopefully easy to visually interpret and also included x-axis ticks as recommended by Referee #1.*

L211 negative not positive

*Thank you for noting this error.*

In the highly skewed distribution (Fig 11), it is getting obvious that the SE does not represent well the uncertainties. Consider reporting quantiles of the distribution which then reflect the higher uncertainties towards higher CH4 flux values.

*We feel that showing the full probability distribution is the most accurate way of demonstrating the range of values. One might argue that a box and whisker or violin plot may be more appropriate for Figure 7, and we would be inclined to agree, but such a plot would be too busy for the human eye to easily render. We also did not want to burden every value placed with maxima, minima, ranges, and the like and we further point out that we were careful to ensure that negative flux values remained in our per-bison flux estimates, rather than thresholding the values at zero, which can bias the full uncertainty distribution of the observations.*

It would be useful and interesting to repeat the measurements with the fodder source placed in the major footprint area.

*Bison and the flux footprint both tended to reside in the south, southwest, and west ends of the pasture. This is a major reason why we chose the particular experimental design. It would be an interesting additional experiment to place feed within the footprint, but this might amount to flux chasing. Bison are powerful and unpredictable animals and entering their enclosure would be very risky (and certainly not allowed by the University). Fodder was delivered by the employees of the landowner over the fence from a safe distance.*

From Fig 3 and Fig 6 it becomes clear how little overlap there is between bison presence in the footprint. How would the flux estimates look like if you just choose the occasions when the joint

presence of many bisons overlaps with the core (i.e. 50% flux contribution) footprint area for a certain time? Such an analysis could enhance the understanding of how robust your estimate is.

*Figure 3 represents a half-hour period and Figure 6 the aggregated flux footprint, which lies predominantly to the southwest. Bison tended to aggregate to the west such that there was considerable overlap between bison and footprint distributions. We do not know how this conclusion was arrived at given that the footprint and bison favored the areas west, southwest, and south of the tower. We recreated the figures to demonstrate the overlap between bison and footprint given that we carefully designed the experiment to ensure reasonable overlap between the footprint and bison distributions.*

Discussion
Give an approximate estimate of the bulk uncertainties inherent to the flux calculations in the discussion section.

*We feel that we did this in the opening paragraph of the Discussion.*

It remains unclear if the low CH4 fluxes for bison fluxes is a result of methodology (spatial distribution, flux footprint uncertainty, non-stationary conditions) and possibly (but probably of much less importance) also other confounding factors (fodder composition).

*We agree but could not test bison methane efflux with respect to diet directly without a calorimeter (and permission from the landowner and University to make such a measurement, neither of which would be likely to be granted and further the animal may have to be sedated and/or at a lower metabolic state to be in a box, resulting in measurement bias). We suspect that a major reason for low methane fluxes is due to energy conservation during winter and hope to confirm this by securing grant funding for a larger study to do so. The seasonal cycle of cattle methane efflux is apparent in Prajapati and Santos (2019) and other references who often assume that the seasonal variability may be due to changes in background sources in their feedlot system. We are curious to know how seasonal metabolic effort impacts $CH_4$ efflux and expanded the discussion of this topic in the revised manuscript.*

In the discussion, it is necessary to more specifically elaborate on why bison CH4 emissions should be that low, what can be reasons/mechanisms behind it?

*We were hesitant to speculate on the reasons for the relatively low per-animal methane efflux but do note that they are rather similar to Prajapati and Santos (2019) and other values from cattle in winter. We note this more explicitly in the revised manuscript. As noted in the above comment we suspect that wintertime energy conservation is a dominant reason and we are interested in exploring seasonal variability in methane efflux further.*

The methodological issues seem to dominate the outcome of the paper and I lack of confidence in the estimated uncertainty.

*We disagree. We treated methodological challenges with an abundance of caution and state this extensively in the text. We included two sensitivity analyses with respect to the footprint analysis*

*that is now extended to include an independent footprint model. We feel that this exceeds the uncertainty analyses of most eddy covariance-based studies.*

---

## Referee Report (RR1)

**Review of manuscript bg-2020-38-manuscript-version4**

This is my first review of an already revised manuscript by Stoy et al. The authors present about 2.5 months of EC flux measurements for methane over a snow-covered pasture field with a managed bison herd. By combining the camera image derived distribution of the animals with EC footprint models the average daily emission rate per animal was derived. The study adopts methods already presented in previous studies (Felber et al., 2015; Dumortier et al., 2017; 2019) with the new element of using camera images to visually determine the position of the animals on the pasture field.

Since, according to the authors, the study presents the first actual measurement of CH4 emissions by bison, the results are valuable despite the limited time period.

However, the manuscript presently suffers from a number of issues that need major revisions before the manuscript is suitable for publication. They are listed in the following comments.

**MAJOR COMMENTS**

1) line 24-25 (and throughout manuscript): I find it generally useful to apply two different footprint models in such a study. However, the animal emission results obtained with the two models should not be treated separately throughout the manuscript and in the abstract. This is not very useful (or even confusing) for the reader, unless the authors want to focus on the specific model differences in detail (which is in my understanding not the scope of this study). Instead, the authors should treat the difference induced by the two models as a part of the footprint model uncertainty. Thus, they should either take the average result of the two models as best guess or declare the preference of one of the models and use only that for the final results.

2) line 138-139: This statement is too optimistic. Heidbach et al. (2017) found considerable differences between different footprint models. For a feedlot experiment, Prajapati and Santos (2018) also found large differences in the spatial extension of footprint models (as mentioned later in lines 326ff.). Therefore, the (systematic) uncertainty effect of the footprint calculation seems to be underestimated in this study (see also comments 3 and 5), although the authors claim a conservative uncertainty estimation.

3) line 146-147: The calculation and use of individual ("unique") $z_0$ values for each half hour is problematic in my view. Such $z_0$ values can vary a lot (even if the roughness conditions in the footprint remains unchanged) especially at low wind speeds. I would therefore like to see the obtained roughness length values in this study and eventually recommend to constrain them to a plausible range. Otherwise, the uncertainty of the footprints may be much larger than expected.

The effect of grazing animals in the flux footprint on the effective roughness length $z_0$ has been analysed e.g. by Felber et al. (2015). It would be useful to compare those results with the findings in the present study.

4) line 160-170: The concept description of the footprint approach is not fully appropriate and unnecessarily complicated.

  a) Eq. 4 is in fact the generic definition (in a discretized math. form) of the footprint weight function $\phi_{ij}$ as presented e.g. by Schmid (1997). It is valid for all EC measurements.

  b) Eq. 5 is unnecessary and complicates the concept presentation. It would be much better to introduce here the (simplifying) assumption of equal average flux per bison $<f_{ij}>$ in the following way: $F_{ij} = n_{ij} <f_{ij}>$.

Inserted in Eq. 4 this directly leads to Eq. 6.

5) line 214-215: Obviously the methane emission by the bison was assumed to be a ground source in the footprint models (i.e. the snow surface in the present experiment). This is clearly questionable since the mouth/nose of a bison (main source) is not generally at ground level. I would assume an average height of 30 to 50 cm. In addition, the exhaled air has an upward inertia due to its high temperature (especially in winter) that leads to an even higher effective source height for the model. It needs to be discussed (or tested) what error/uncertainty is introduced by a wrong emission height.

6) line 260ff. The discussion of the obtained results in comparison to existing literature information is unsystematic and clearly insufficient. The methane emission of bovine animals mainly depends on the amount of feed intake and its (digestible) energy content (see line 64-65). The feed intake depends on the energy demand of the animal, which is itself a function of the body weight and the productivity (milk yield, weight gain). This has to be taken into account when discussing the different $CH_4$ emissions by different animals in other studies. It could be checked whether existing functional relationships for bovine animals (depending on animal weight and feed amount and characteristics, as given in Table S1-S3) could be used to calculate emissions for comparison with the EC derived results.
The authors cited Kelliher and Clark (2010), Smith et al. (2016) and Hristov (2012) in the introduction (line 59-63) to point out the importance of bison $CH_4$ emission in pre-industrial times. These authors obviously use estimates of typical bison $CH_4$ emissions. Why are those results not included in the discussion in comparison to the results of the present study.

7) line 281-309: This section has a misleading title because it only discusses fluxes with bison absent from the pasture field. I also consider this part as quite speculative because it is based on a small dataset. Given that the soil/surface methane fluxes are negligible in comparison to the animal enteric fermentation emissions (Fig. 8), this section is not contributing significantly to the aims of the study and should be shortened drastically or should be fully omitted. In turn, the discussion of entheric $CH_4$ emission needs to be expanded (see previous comment).

8) Figure 3: This graph is not useful to provide a (quantitative) information of a typical flux footprint extension. It should be changed to a contour plot with contour lines enclosing areas of 50%, 70%, 90% contribution to the flux (as commonly provided in similar studies).
Also the u* and z/L values of the displayed example should be indicated in the figure caption.

9) It would be useful to add a (short) specific Conclusions section

MINOR COMMENTS

line 25: "...similar to eddy covariance measurements of methane efflux from a cattle feedlot during winter". This is a very unclear statement. I suggest to clarify e.g. to: "...similar to previously reported eddy covariance derived cattle methane emissions in a feedlot during winter".

line 47-49: The message of this sentence is misleading. When comparing bison to other ruminants in agricultural production, the differences in productivity (rate of weight increase) and the general energy demand of the animal is much more important for the $CH_4$ emission than some grazing preference details.

line 52-55: It appears not logical that the authors first cite studies from 2013 and 2017 and afterwards state "Recent studies have revised methane emission estimates from livestock upward by over 10%" with citing of mainly older (!) studies. Please rephrase.

line 185: "perfectly aggregated" sounds strange in this context (because the animals need to be distant to each other, i.e. non-aggregated). I suggest to change to "perfectly distributed".

line 188: "...the true number of each bison..." is unclear. Moreover, what is the difference between the true number and the measured number? Please rephrase and clarify.

line 324-329: This is a quite unspecific discussion of the problem without a clear conclusion concerning the effect of footprint uncertainty (see comment 2 above).

line 338-340: It should be mentioned here, that eddy covariance might not be suitable to determine (separate) the efflux of individual animals on the pasture even if they can be identified and tracked (especially because they tend to move in groups).

Figure 7 caption: Change to "...from the study pasture near Gallatin Gateway ..."

ADDITIONAL REFERENCES

Prajapati, P., and E.A. Santos. 2018. Estimating methane emissions from beef cattle in a feedlot using the eddy covariance technique and footprint analysis. Agric. For. Meteorol. 258:18–28. doi:10.1016/j.agrformet.2017.08.004

Schmid, H.P. 1997. Experimental design for flux measurements: matching scales of observations and fluxes. Agric. For. Meteorol., 87, 179–200.

---

## Referee Report (RR2)

I really appreciate the reading of this paper which is well written. The application of eddy covariance to wild animals is innovative and interesting. The method is well described and the obtained results are very clear. However, I have the following major remarks:

- I do not see the point of using the Tikhonov Regularization method. If cattle location were biased, their location could more or less aggregated than observed. Moreover, cattle could be present in a pixel different than the one expected. I do not understand why the spatial uncertainty should be estimated by smoothing cattle distribution. In my opinion, a better way to estimate spatial uncertainty would be to consider that some part of the herd might not be in the expected cell but in an adjacent cell, even if it results in a more aggregated cattle distribution (e.g. 10% of the herd is considered as mislocated in a specific direction, one iteration could be done for each cardinal direction). As this element represent an important part of the manuscript I suggest either to better explain why the proposed option is the good one (which I am not convinced) or to propose another spatial uncertainty estimation method.
- There is a problem with Equation 6 which should be written as:

$$\langle f_x \rangle = \frac{F_x}{\sum_{i=1}^{8} \sum_{j=1}^{12} n_{ij} \phi_{ij}}$$

$f_x$ corresponds to mol animal$^{-1}$ s$^{-1}$

$F_x$ corresponds to mol m$^{-2}$ s$^{-1}$

$n_{ij}$ corresponds to an amount of animals

$\phi_{ij}$ corresponds to m$^{-2}$

The equation proposed in the manuscript is not homogeneous. I hope that the equation which was applied in the calculations was the one described here above and not the one described in the manuscript.

I also have the following minor remarks:

-line 27 to 28: wording: "Our observations point to the need for direct comparisons of methane emissions from conventional and alternate grazing systems using eddy covariance": I do not see the link between observations and grazing management.
-line 43: wording: "also need not migrat

-line 44: The fact that bison do not follow the "green wave" and that they tend to stimulate plant growth does not suggest that they select for forage quality rather than quantity.

-line 86 / Table S3: the average mass of a bail should be specified.

-Equation 2 /3: is it an "$\alpha$" (Equation 2) or a "a" (Equation 3).

-line 159:I would say: "are solely due to"

-line 216: Giving the mean and median wind direction is of limited interest (e.i. if there are 2 main wind directions, the mean wind direction will be in-between, in a direction from which the wind might never be coming from). Main wind directions are far more relevant.

-line 243: Wording: I would say 'At least one bison was located within…"

-line 244: which

-line 244 to 245: wording: "an average of 8 models which increased to both footprint models"?

-line 265-278: In my opinion, the discussion would be more interesting if emission comparisons would consider animal body weight => e.g. comparisons in $kg_{CH4}kg_{bw}^{-1}day^{-1}$

-line 301: Gourlez de la Motte

-line 303: Results from Dengel indicate that CH4 fluxes are more important during summer but the main reason is a higher stocking density on the pasture. Manure impact on this result is expected to be weak / negligible as manure is not placed in anaerobic conditions. I do not think that the publication from Dengel support the associated sentence.

-line 311: Gourlez de la Motte

-line 345 : wording : « algorithms for are »

-line 351: remove "and"

-Figure 11: the second "Figure 11" should be named "Figure 12".

---

## Referee Report (RR3)

**Review of manuscript bg-2020-38-manuscript-version5**

This is my second review of the manuscript and the third review round in total.
I am not satisfied with the revisions and the response of the authors to my previous comments. The authors decided to rebut most of my major comments and their response contains, in my view, partly insufficient or inappropriate arguments.
In the following I list the remaining issues and requested improvements.

MAJOR COMMENTS

A) The uncertainty of the main result, the average per-bison-emission, is not treated appropriately. In the new manuscript version there is even a new aspect on this issue.
The authors give the results as "mean ± standard deviation" (e.g. abstract, line 249, ...). It is unclear what that means (standard deviation of what dataset?). Also a standard deviation is usually not a useful uncertainty measure. This needs to be clarified.
- In response to my previous comments, the authors have added some text statements about additional uncertainty sources, but at the same time, they apparently have reduced the uncertainty estimation, instead of increasing it.
- line 198: The 17% uncertainty "for longterm sums" adopted from the literature can hardly be used for the present extremely non-homogenous situation and a very limited measurement time of only daytime cases during only about 3 winter weeks (bison present and camera pics available).
- It is also crucial to declare in the manuscript, how many half-hourly per-bison-emission values were available after all quality filters.
- Figure 11A shows that the individual per-bison-emission data have a strongly skewed distribution. Therefore the random-like error cannot be well estimated according to Gaussian statistics rules. However the difference between arithmetic mean and median is an indicator of a large uncertainty.

B) I consider the discussion of the per-bison-emission results in comparison to the literature as still insufficient. In the response to my previous comment 6, the authors state that "Methane flux is related to the animal in question, its body mass, diet, metabolic state, pregnancy / weaning status, and more." But for the literature comparison they just selected "...results that are similar to ours". This is a clearly non-scientific approach. The authors should not just select literature per-animal emission values that are similar to the present study without considering/stating the relevant factors (body mass, diet, etc.) in the referenced studies.
Since the authors compare their results to feedlot studies, it is also not clear whether the present experiment is considered as representative/comparable to a grazed pasture system or rather to a feedlot system. This should be clarified.

MINOR AND LANGUAGE COMMENTS

C) Response to prev. comment 3:
Since the authors introduce their z0 determination in detail in Section 2.4 and because the z0 values are important inputs for the footprint models, it is surly warranted that the authors present some corresponding results in the text (e.g. range of obtained z0 values with/without animals in the footprint).

D) Response to prev. comment 9:
The rebuttal indicate that the authors principally put the use of a 'Conclusions' section into question, because their arguments could be applied to most scientific papers. I do not agree but I leave it to the editor to decide this issue.

E) line 22: "...similar to eddy covariance measurements of methane efflux from a cattle feedlot during winter". It needs to be clarified here, that the mentioned feedlot results were not obtained in this study but that you mean "previously reported eddy covariance measurements ..."

F) line 163: define the meaning of $F_{ij}$ in Eq. 4

G) line 321: It needs to be clarified whether the 36% is a attribution fraction or a relative uncertainty.

H) Figure 10 and 11A: The use of the term "probability" is strongly misleading here. The shown data rather are observed frequencies of occurrence.

---

## Author Response (AR2)

your revised manuscript "Methane efflux from an American bison herd" has been seen by two independent reviewers. Both reviewers think that the study is innovative, interesting and valuable despite the relatively short time period of observations.
Yet, both reviewers are also critical, with reviewer 1 questioning the approach taken to assess spatial uncertainty and reviewer 2 being particularly critical about the use of the two footprint models and the assumption on where methane is emitted from ie difference in height between snow surface vs bison head resulting in a major revisions suggestion. Given the additional comments made, the paper is not yet ready for publication.
While this is the second round of reviews, both reviewers as well as myself share the opinion of this being a valuable study, Thus, I ask you to address the issues raised by clarifying and possible simplifying the manuscript as well as by removing the speculative bits.

with kind regards

Lutz Merbold
Referee #3

We thank the Referees for their insightful comments. They raise a number of important technical points, many of which were particularly welcome because they helped simplify the analysis. We detail our responses to each point below. Thank you for your continued support of this manuscript and we hope that we have addressed the Referees' comments adequately.

Review of manuscript bg-2020-38-manuscript-version4

This is my first review of an already revised manuscript by Stoy et al. The authors present about 2.5 months of EC flux measurements for methane over a snow-covered pasture field with a managed bison herd. By combining the camera image derived distribution of the animals with EC footprint models the average daily emission rate per animal was derived. The study adopts methods already presented in previous studies (Felber et al., 2015; Dumortier et al., 2017; 2019) with the new element of using camera images to visually determine the position of the animals on the pasture field.

Since, according to the authors, the study presents the first actual measurement of CH4 emissions by bison, the results are valuable despite the limited time period.
However, the manuscript presently suffers from a number of issues that need major revisions before the manuscript is suitable for publication. They are listed in the following comments.

Thank you for your insight, you have raised a number of points that have been on our mind. We hope that we have adequately addressed these in the responses below.

MAJOR COMMENTS

1) line 24-25 (and throughout manuscript): I find it generally useful to apply two different footprint models in such a study. However, the animal emission results obtained with the two

models should not be treated separately throughout the manuscript and in the abstract. This is not very useful (or even confusing) for the reader, unless the authors want to focus on the specific model differences in detail (which is in my understanding not the scope of this study). Instead, the authors should treat the difference induced by the two models as a part of the footprint model uncertainty. Thus, they should either take the average result of the two models as best guess or declare the preference of one of the models and use only that for the final results.

We agree with this comment; we added both footprint models in response to previous reviews and agree that continually noting them both is excessive. Instead we take your advice to focus on the average of both and to use their differences to gain insight into uncertainties due to choice of footprint model. We retain many of the figures that detail the different behavior of the models as we feel that it helps provide clarity as to why they are different.

2) line 138-139: This statement is too optimistic. Heidbach et al. (2017) found considerable differences between different footprint models. For a feedlot experiment, Prajapati and Santos (2018) also found large differences in the spatial extension of footprint models (as mentioned later in lines 326ff.). Therefore, the (systematic) uncertainty effect of the footprint calculation seems to be underestimated in this study (see also comments 3 and 5), although the authors claim a conservative uncertainty estimation.

Footprint model accuracy is an ongoing challenge in flux science. The purpose of this experiment is not to compare footprint models given that measurements were made under field conditions at a working ranch with non-domesticated animals rather than an experimentally controlled setting. Rather than add additional footprint models, we revised the language of the manuscript to note that uncertainties due to footprint modeling may be larger than we estimate using the two models chosen here. We then further emphasize that robust approaches to estimating footprint uncertainty remain a critical area of future research.

3) line 146-147: The calculation and use of individual ("unique") $z_0$ values for each half hour is problematic in my view. Such $z_0$ values can vary a lot (even if the roughness conditions in the footprint remains unchanged) especially at low wind speeds. I would therefore like to see the obtained roughness length values in this study and eventually recommend to constrain them to a plausible range. Otherwise, the uncertainty of the footprints may be much larger than expected.

The effect of grazing animals in the flux footprint on the effective roughness length $z_0$ has been analysed e.g. by Felber et al. (2015). It would be useful to compare those results with the findings in the present study.

We somewhat disagree with this comment because we feel that animal movement results in a situation where $z_0$ is dynamic across time. The approach of Felber et al. (2015) for constraining the likely range of $z_0$ is interesting and they are correct in noting that it should take a reasonably constrained seasonal course as a function of grass height in their case. In our conditions with snow and obstacles (animals), we argue that it makes sense to consider $z_0$ as a dynamic variable because it can change rapidly. We did in retrospect take elements of the approach of Felber et al. (2015) by thresholding the extreme values that result when the denominator of the equation for $z_0$ approach zero. This thresholding only impacted 15 of 3117 observations.

4) line 160-170: The concept description of the footprint approach is not fully appropriate and unnecessarily complicated.

a) Eq. 4 is in fact the generic definition (in a discretized math. form) of the footprint weight function $\phi_{ij}$ as presented e.g. by Schmid (1997). It is valid for all EC measurements.

This is correct. We are starting with the basics. This equation was also presented by Dumortier et al. (2019) as noted and we adjusted it for the dimensions of our field. We added the Schmid (1997) reference for completeness.

b) Eq. 5 is unnecessary and complicates the concept presentation. It would be much better to introduce here the (simplifying) assumption of equal average flux per bison $<f_{ij}>$ in the following way: $F_{ij} = n_{ij} <f_{ij}>$.
Inserted in Eq. 4 this directly leads to Eq. 6.

We understand what you are saying but we do not feel that equation 5 is unnecessary because it is rather simple and defines $f_X$, used in the next equation. It is taken from Dumortier et al. (2019) as indicated, which was the inspiration for the footprint approach, where it also happens to be their equation 5. We note also in response to Referee #4 that we did erroneously list the normalization term $\Delta x_{ij}\Delta y_{ij}$ twice in equations 4 and 6. This has been corrected in the revised manuscript and did not impact the results.

5) line 214-215: Obviously the methane emission by the bison was assumed to be a ground source in the footprint models (i.e. the snow surface in the present experiment). This is clearly questionable since the mouth/nose of a bison (main source) is not generally at ground level. I would assume an average height of 30 to 50 cm. In addition, the exhaled air has an upward inertia due to its high temperature (especially in winter) that leads to an even higher effective source height for the model. It needs to be discussed (or tested) what error/uncertainty is introduced by a wrong emission height.

This is a topic that we discussed when preparing the paper. As it happens, measured snow depth was nearly 30 cm across much of the measurement period, noting that snow depth across the entire field becomes rather dynamic due to drifting and trampling. Per the latter point, when bison are grazing, which is often, their mouths are *underneath* the snow surface. We agree that a brief discussion of methane release heights is warranted and have added it to the text, but we have little basis to adjust release height up and down for the dynamic vertical head movements of multiple animals so we did not pursue this as we felt that it would add uncertainty.

The second point regarding temperature differences also came to mind when designing the experiment. Methane leaves the animal at its body temperature, which almost always differs from air temperature. Heat is also transferred more efficiently than passive scalars in the convective sublayer (e.g. Katul et al., 1995, DOI: 10.1007/BF00712120) so we cannot assume that methane is equally buoyant to heat. This brings to mind the need to explore heated point release experiments and footprint models to quantify the importance of heated air parcels in biogeochemical studies, and we now note both of these topics in the revised Discussion section.

6) line 260ff. The discussion of the obtained results in comparison to existing literature information is unsystematic and clearly insufficient. The methane emission of bovine animals mainly depends on the amount of feed intake and its (digestible) energy content (see line 64- 65). The feed intake depends on the energy demand of the animal, which is itself a function of the body weight and the productivity (milk yield, weight gain). This has to be taken into account when discussing the different CH4 emissions by different animals in other studies.
It could be checked whether existing functional relationships for bovine animals (depending on animal weight and feed amount and characteristics, as given in Table S1-S3) could be used to calculate emissions for comparison with the EC derived results.
The authors cited Kelliher and Clark (2010), Smith et al. (2016) and Hristov (2012) in the introduction (line 59-63) to point out the importance of bison CH4 emission in pre-industrial times. These authors obviously use estimates of typical bison CH4 emissions. Why are those results not included in the discussion in comparison to the results of the present study.

Previous studies noted a rather wide range of methane emission results from different animals as a function of food quality and quantity. Methane flux is related to the animal in question, its body mass, diet, metabolic state, pregnancy / weaning status, and more. Any universal model for methane flux – and we are aware of none – must account for all of these variables.

It would be interesting to create such a model, but this exceeds the scope of the present study. We did check to see if the different nutrient content of the different hay bails (Table S2) impacted methane flux and found evidence that they did at the $P < 0.1$ level but not the more commonly used $P < 0.05$. We did not subsequently dwell on this result because we were not interested in reporting results of questionable significance. In lieu of a comprehensive analysis of all previously published methane flux results, which would be an interesting topic for a review paper, we noted results that are similar to ours. This required a bit of digging; the comparisons to both the Hammond et al. (2016) results and the Prajapati and Santos (2019) results came from a careful analysis of their tables and figures. The purpose of the comparisons is to note that our results have a foundation in the literature.

The Kelliher and Clark (2010) used the Galbraith et al. (1998) penned bison flux estimates and adjusted them for metabolic scalars determined for cattle. The Smith et al. (2016) manuscript uses the Kelliher and Clark estimates, and Hristov (2012) uses a combination of both. Because we questioned the extrapolation of methane flux measurements in a penned animal fed pelletized feed to conditions in the field, we likewise did not belabor a comparison in the Discussion section but rather applaud the efforts of all of the authors for making these difficult measurements and applying them for an improved understanding of the earth system.

7) line 281-309: This section has a misleading title because it only discusses fluxes with bison absent from the pasture field. I also consider this part as quite speculative because it is based on a small dataset. Given that the soil/surface methane fluxes are negligible in comparison to the animal enteric fermentation emissions (Fig. 8), this section is not contributing significantly to the aims of the study and should be shortened drastically or should be fully omitted. In turn, the discussion of entheric CH4 emission needs to be expanded (see previous comment).

We wanted to take care to ensure that methane efflux can be attributed to bison and not other factors in a pasture near a river. We still have hundreds of measurements even though the data record is not as long as some other flux studies. It would have been nice to have a longer data record but we did not wish to alter the bison management strategy of the landowner; the animals were moved to a different pasture at the end of their tenure in the field that we instrumented. We did not leave instrumentation on the field for an extended period after the bison left in order to avoid unnecessary conflict with the landowner. We agree that the title of the subsection could be more accurate and we now write: '4.1 Methane and carbon dioxide efflux in response to environmental variables and bison presence'.

8) Figure 3: This graph is not useful to provide a (quantitative) information of a typical flux footprint extension. It should be changed to a contour plot with contour lines enclosing areas of 50%, 70%, 90% contribution to the flux (as commonly provided in similar studies).
Also the u* and z/L values of the displayed example should be indicated in the figure caption.

We created a figure that exhibits both the Hsieh et al. (2000) and Kljun et al. (2015) footprints for a single half-hourly measurement period, thank you for the suggestion.

9) It would be useful to add a (short) specific Conclusions section

We deliberated this but we summarized findings in the first paragraph of the Discussion and any conclusion would be somewhat redundant with the Abstract. We omitted a Conclusions section for brevity noting that the end of the Discussion section expands the text back to the earth system processes that motivated the original study.

MINOR COMMENTS

line 25: "...similar to eddy covariance measurements of methane efflux from a cattle feedlot during winter". This is a very unclear statement. I suggest to clarify e.g. to: "...similar to previously reported eddy covariance derived cattle methane emissions in a feedlot during winter".

We have to disagree with this point. We feel that the text as written is more efficient and equally if not more accurate because methane emissions in this case will arise from more than cattle themselves; manure will also be a methane source in a feedlot system even if it might be relatively minor depending on how it is managed.

line 47-49: The message of this sentence is misleading. When comparing bison to other ruminants in agricultural production, the differences in productivity (rate of weight increase) and the general energy demand of the animal is much more important for the CH4 emission than some grazing preference details.

We also have to disagree with this point because multiple studies have demonstrated the importance of diet and herd management for ruminant methane emissions, for example Waghorn et al. (2011, 10.1016/j.anifeedsci.2011.04.019). We agree in general that productivity and energy

demand are critical for methane efflux but extensive effort is underway to change ruminant feed to minimize methane losses, which demonstrates the importance of feed.

line 52-55: It appears not logical that the authors first cite studies from 2013 and 2017 and afterwards state "Recent studies have revised methane emission estimates from livestock upward by over 10%" with citing of mainly older (!) studies. Please rephrase.

Measurements were made in 2017 and 2018 so the 2017 reference was quite new at the time and the 2013 reference was not exactly older. We revised the statement to read 'Methane emission estimates from livestock have tended to increase as more information becomes available'.

line 185: "perfectly aggregated" sounds strange in this context (because the animals need to be distant to each other, i.e. non-aggregated). I suggest to change to "perfectly distributed".

We agree that the suggestion makes the text more clear and changed it.

line 188: "...the true number of each bison..." is unclear. Moreover, what is the difference between the true number and the measured number? Please rephrase and clarify.

All measurements have uncertainty. We reworded the text to state 'All observations have uncertainty so estimates of bison location using cameras provides an initial guess of the true location.' In the revised manuscript we now use camera observations as an initial guess of the bison location and add a stochastic uncertainty analysis.

line 324-329: This is a quite unspecific discussion of the problem without a clear conclusion concerning the effect of footprint uncertainty (see comment 2 above).

We do not feel that the issue of footprint uncertainty has been adequately addressed by the community; adding more footprint models will likely lead to more uncertainty

line 338-340: It should be mentioned here, that eddy covariance might not be suitable to determine (separate) the efflux of individual animals on the pasture even if they can be identified and tracked (especially because they tend to move in groups).

This is an interesting point and we agree. We added 'That being said it will be difficult to measure the methane contributions of individual animals in species that tend to herd using eddy covariance'.

Figure 7 caption: Change to "...from the study pasture near Gallatin Gateway ..."

We like this suggestion and have changed the text.

ADDITIONAL REFERENCES

Prajapati, P., and E.A. Santos. 2018. Estimating methane emissions from beef cattle in a feedlot using the eddy covariance technique and footprint analysis. Agric. For. Meteorol. 258:18–28. doi:10.1016/j.agrformet.2017.08.004

Schmid, H.P. 1997. Experimental design for flux measurements: matching scales of observations and fluxes. Agric. For. Meteorol., 87, 179–200.

Referee #4
I really appreciate the reading of this paper which is well written. The application of eddy covariance to wild animals is innovative and interesting. The method is well described and the obtained results are very clear. However, I have the following major remarks:
I do not see the point of using the Tikhonov Regularization method. If cattle location were biased, their location could more or less aggregated than observed. Moreover, cattle could be present in a pixel different than the one expected. I do not understand why the spatial uncertainty should be estimated by smoothing cattle distribution. In my opinion, a better way to estimate spatial uncertainty would be to consider that some part of the herd might not be in the expected cell but in an adjacent cell, even if it results in a more aggregated cattle distribution (e.g. 10% of the herd is considered as mislocated in a specific direction, one iteration could be done for each cardinal direction). As this element represent an important part of the manuscript I suggest either to better explain why the proposed option is the good one (which I am not convinced) or to propose another spatial uncertainty estimation method.

Thank you for the kind comments and insightful review. In an earlier version of the manuscript we included a table that presented average per-bison methane flux estimates after shifting their locations relative to the footprint in the four cardinal directions. Previous referees deemed this unpalatable so we did away with it.

We chose the Tikhonov Regularization method as a sensitivity analysis in part because it does its effects can be visualized on a single axis and it not include a stochastic component that would require Monte Carlo methods. We agree with you that it would be interesting to explore other methods and this comment made us rethink the best way of exploring uncertainty due to spatial location.

A random distribution of locations across space follows a Poisson point process. But our pictures reveal more about bison locations than a random draw. Instead, we now use the bison observations as an initial guess, use Tikhonov Regularization to simulate likelihood surfaces, and then simulate a distribution of bison that uses observations as a prior but adds a stochastic component to bison movement within half-hour periods. We feel that maintaining the Tikhonov Regularization is important to give a finite likelihood for the Poisson point process to allocate bison to pixels adjacent to where they were estimated. We now present the original sensitivity analysis Tikhonov Regularization in a new Supplemental Information section.

There is a problem with Equation 6 which should be written as:
$\langle f_x \rangle = F_x / \left( \sum_{i=1}^{8} \sum_{j=1}^{12} [\![ n_{ij} \, \phi_{ij} ]\!] \right)$
$f_x$ corresponds to mol animal-1 s-1

$F\_x$ corresponds to mol m-2 s-1
$n\_ij$ corresponds to an amount of animals
$\phi\_ij$ corresponds to m-2
The equation proposed in the manuscript is not homogeneous. I hope that the equation which
was applied in the calculations was the one described here above and not the one
described in the manuscript.

Thank you for noticing this, we accidentally wrote the scaling factor $\Delta x\_ij \; \Delta y\_ij$ twice in both equations 4 and 6 in the text but not in the code used for the calculations. We corrected the text.

I also have the following minor remarks:
-line 27 to 28: wording: "Our observations point to the need for direct comparisons of methane emissions from conventional and alternate grazing systems using eddy covariance": I do not see the link between observations and grazing management.

Bison grazing typically replaces cattle grazing systems in rangelands. We re-worded the last sentence of the abstract in response to this and other comments.

-line 43: wording: "also need not migrat
-line 44: The fact that bison do not follow the "green wave" and that they tend to stimulate plant growth does not suggest that they select for forage quality rather than quantity.

We understand how this was confusing on lines 43-44; stimulating plant growth implies that nutrient-rich young leaves are being eaten but this is indirect. Instead we simplified the text to read: They tend to graze in preferred meadows during winter and search broadly for the most energy-dense forages during the growing season (Fortin et al., 2003 Geremia et al., 2019), often in areas which have recently burned (Allred et al., 1991; Coppedge and Shaw, 1998; Vinton et al., 1993).

-line 86 / Table S3: the average mass of a bail should be specified.

Thank you for pointing this out, we have added average bail mass to the table legend.

-Equation 2 /3: is it an "α" (Equation 2) or a "a" (Equation 3).

The equation should have *a* because α is used in Equation 7. Thank you for noticing this error.

-line 159:I would say: "are solely due to"

We feel that 'solely' is probably too strong a word as there are certainly minor sources and sinks distributed throughout the field that average to approximately zero.

-line 216: Giving the mean and median wind direction is of limited interest (e.i. if there are 2 main wind directions, the mean wind direction will be in-between, in a direction from which the wind might never be coming from). Main wind directions are far more relevant.

We agree in principle and the previous version of the manuscript included a histogram of wind directions. Because the wind rose shows a predominant wind direction (Figure 6) we feel that discussing the mean is accurate in our case.

-line 243: Wording: I would say 'At least one bison was located within…"
-line 244: which
-line 244 to 245: wording: "an average of 8 models which increased to both footprint models"?

We revised the wording in these passages and also revised wording throughout the manuscript for clarity when we felt it necessary.

-line 265-278: In my opinion, the discussion would be more interesting if emission comparisons would consider animal body weight => e.g. comparisons in 〖kg〗_CH4 〖kg〗_bw^(-1) 〖day〗^(-1)

We hesitated to perform this analysis because of how the animals tended to cluster with younger animals often nearer larger animals and larger animals often roaming more throughout the pasture, i.e. smaller animals were infrequently alone. Our approach did not allow us to track animals individually but we agree with the comment that it would be interesting to perform such an analysis if it could be done without introducing potential biases.

-line 301: Gourlez de la Motte

Thank you for instructing us on the correct usage here and on line 311 below.

-line 303: Results from Dengel indicate that CH4 fluxes are more important during summer but the main reason is a higher stocking density on the pasture. Manure impact on this result is expected to be weak / negligible as manure is not placed in anaerobic conditions. I do not think that the publication from Dengel support the associated sentence.

We revised the sentence to note that manure may be a methane source but is likely rather low in rangeland systems for which we now cite Steed Jr. and Hashimoto (1994).

-line 311: Gourlez de la Motte

(see above)

-line 345 : wording : « algorithms for are »

We corrected the passage, thank you for noticing this.

-line 351: remove "and"

We carefully read the passage and are not sure what this note refers to, but we should note that we comprehensively reviewed the manuscript for minor usage errors.

-Figure 11: the second "Figure 11" should be named "Figure 12".

This is correct, we made a clerical error when rearranging figures. The figure legends and their references in the text are now corrected.

[revised manuscript text omitted]

---

## Author Response (AR3)

Dear Paul Stoy et al.,

I have now received the third review for your revised manuscript on methane emissions from bison. While the reviewer is not completely convinced by the responses you have given in your previous response I am confident the currently remaining issues can be clarified, particularly in terms of explaining which errors you are showing based on how many data points and how these were derived. Similarly, I suggest to address the importance of specific biological parameters on CH4 emissions from ruminants (ie. diet quality not just quantity). Of course, there aren't many studies on bison available and this seems to be a scoping study to extend future measurements, I would like to point towards comparing your results to either "similar" species such as buffalo or cattle that are grazed even during colder periods in the year. You may further think about stressing more strongly that your system is a mix between a grazed and a feedlot system, given the supplementation with hay.
With this, I am looking forward to receiving your revised manuscript.

with kind regards

Lutz Merbold

*Dear Dr. Merbold,*
*Thank you for reconsidering the manuscript. We made a number of additional changes to the manuscript which we believe improved the analysis and its presentation, and also added a short Conclusions section to the Discussion and a Land Acknowledgement to the Acknowledgements section. We added new references on ruminant nutrition and methane efflux and revised the Discussion section to explain our comparison with other literature estimates and to frame our results in a broader context. We thank you for your support of the manuscript and please do not hesitate to write if questions arise; we would be happy to further reconsider any analysis and were relieved that our efforts in comprehensively re-analyzing all values did not result in meaningful changes to our original study.*

*Sincerely,*
*Paul C. Stoy*

Review of manuscript bg-2020-38-manuscript-version5

This is my second review of the manuscript and the third review round in total. I am not satisfied with the revisions and the response of the authors to my previous comments. The authors decided to rebut most of my major comments and their response contains, in my view, partly insufficient or inappropriate arguments. In the following I list the remaining issues and requested improvements.

*We have endeavored to make the suggested changes and are grateful for the suggestions which improved the manuscript. We disagreed with some points from the previous review that we*

*have now added to the manuscript for completeness. One point regards the Conclusions section which we argued is a matter of preference. We added a succinct conclusions section that highlights the key findings of the manuscript and agree that it helps frame the analysis. We also now adopt the Referee's suggestion change the discussion of the equations while still ensuring that we credited the studies that inspired our approach. Another important point that was raised before is the issue source height. Source height would have been interesting to include if we had a basis for determining it, but we had no observational basis for doing so and felt that it would add unnecessary speculation into the analysis. This is one of the most cautious flux studies that we are aware of and we are trying to be very careful to not introduce unnecessary uncertainties. The third regards the z0 calculation. We felt that it would be inaccurate to change our approach, which was designed for the specifics of the study field and measurement period. We do note that the median z0 value that we arrive at with bison,*

MAJOR COMMENTS
The uncertainty of the main result, the average per-bison-emission, is not treated appropriately. In the new manuscript version there is even a new aspect on this issue. The authors give the results as "mean ± standard deviation" (e.g. abstract, line 249, ...). It is unclear what that means (standard deviation of what dataset?). Also a standard deviation is usually not a useful uncertainty measure. This needs to be clarified.

*We summed variances calculated from multiple independent sources of uncertainty. We agree with the Referee that we did not communicate uncertainty correctly for two reasons. One is that we maintained both mean and median values after propagating uncertainty. We should have only kept the mean because large values are 'averaged out' to a degree. The non-normal distribution of methane flux observations that we show in Fig. 11a are sensible in our opinion because we expect intermittent large pulses of methane efflux. It is common to sum observations from non-normally-distributed eddy covariance observations to arrive at temporally-summed observations.*

*In response, we decided to retain standard deviation when presenting the average fluxes with and without bison because we felt that presenting observations in this way is actually more conservative. Had we used variance, for example, the average methane flux without bison would be −0.0009 ± 6.4e-5 $\mu mol\ m^{-2}\ s^{-1}$ and had we used standard error of the mean this value would be −0.0009 ± 0.0002307 $\mu mol\ m^{-2}\ s^{-1}$. Needless to say we are more than happy to present values in this way if you prefer.*

*We re-ran all analyses (at a relatively large amount of computer time given our stochastic approach for estimating bison location uncertainty) and feel that our results are accurate but as the Referee notes should have been presented more clearly. We have revised the discussion of mean fluxes and their sums as a result to be even more conservative with our analysis as described in more detail below.*

- In response to my previous comments, the authors have added some text statements about additional uncertainty sources, but at the same time, they apparently have reduced the uncertainty estimation, instead of increasing it.

*This is correct. Uncertainty need not monotonically increase every time that it is re-evaluated.*

- line 198: The 17% uncertainty "for longterm sums" adopted from the literature can hardly be used for the present extremely non-homogenous situation and a very limited measurement time of only daytime cases during only about 3 winter weeks (bison present and camera pics available).

*Random eddy covariance uncertainty is usually assumed to be on the order of 10-15% for carbon dioxide fluxes following for example Goulden et al., 1996, doi: 10.1111/j.1365-2486.1996.tb00070.x. The uncertainty of flux sums is smaller than individual measurements because random error is averaged out. We had used 17% as the most conservative value for flux sums that was supported by the literature, and we agree that 'long-term' is subjective and removed it. In the revised analysis we use the 41% value from Deventer et al. (2019) to estimate uncertainty due to eddy covariance observations and find that they now make a slightly larger contribution to total uncertainty (25%) but not enough to change our conclusions. To be honest we are more concerned with potential bias uncertainties in all eddy covariance observations than random uncertainties and members of our team have been very active in addressing the eddy covariance energy imbalance challenge and other potential sources of bias uncertainty, which may be important in all flux studies but difficult to quantify.*

- It is also crucial to declare in the manuscript, how many half-hourly per-bison-emission values were available after all quality filters.

*After all filters were applied, 158 half-hourly observations with bison in the flux footprint remained when applying the Hsieh et al. (2000) footprint model and 146 observations were available when applying the Kljun et al. (2015) footprint model. In both instances this is over 100 more measurements than the groundbreaking work of Galbraith et al. (1998) who measured five penned bison with seven replicates. These are also the first field observations made of a species of critical importance to the culture of multiple Indigenous Tribes of North America and their duration was a consequence of respect for the private landowner who allowed us to make these measurements. We now report these values in the manuscript.*

- Figure 11A shows that the individual per-bison-emission data have a strongly skewed distribution. Therefore the random-like error cannot be well estimated according to Gaussian statistics rules. However the difference between arithmetic mean and median is an indicator of a large uncertainty.

*We are unaware of flux observations that follow a normal distribution and the sum of these values will approach a normal distribution due to the Central Limit Theorem. For example, the figure below demonstrates 1000 realizations of the cumulative sum of observed per-bison*

*methane flux with a random 41% uncertainty about each observation, which the upper end of the uncertainty established by Deventer et al. (2019).*

[Figure]

*The cumulative sums, right, are normally distributed. We would agree that our observations are not strictly speaking independent as the central limit theorem requires, but we feel that it is a reasonable approximation.*

*Performing this analysis inspired us to reinterpret the uncertainty analysis using approaches that are perhaps a bit more established in flux science. We calculated the mean percent uncertainty for the spatial uncertainty analysis, the flux footprint analysis (without spatial uncertainty), and the 41% conservative uncertainty estimate of flux sums following Deventer et al. (2019). We used these percent uncertainties to calculate a variance for each observation, then drew 1000 samples from each of these distributions to create different realizations of the sum of fluxes as demonstrated above. We then calculated 95% confidence intervals about the flux sums and present these in the revised manuscript rather than standard deviations or variances. The percent uncertainty that we now calculate for each term differs slightly from previous uncertainty estimates as a result, but results do not change our conclusions.*

B) I consider the discussion of the per-bison-emission results in comparison to the literature as still insufficient. In the response to my previous comment 6, the authors state that "Methane flux is related to the animal in question, its body mass, diet, metabolic state, pregnancy / weaning status, and more." But for the literature comparison they just selected"...results that are similar to ours". This is a clearly non-scientific approach. The authors should not just select literature per-animal emission values that are similar to the present study without considering/stating the relevant factors (body mass, diet, etc.) in the referenced studies.

*We are first and foremost interested in ensuring that our observations are reasonable with respect to other ruminant systems: a simple logic check. We re-worded the text to be more clear on this point. This logic check follows the work of Galbraith et al. (1998) who also noted that bison methane flux are to a first order similar to cattle (at least when fed alfalfa). There are no existing measurements of methane flux from bison in a natural setting and we had to resort to comparisons with non-native introduced cattle grazing systems as a consequence.*

*There are hundreds of studies on cattle and for good reason; they are critical to the global methane cycle but are also favored by European management systems, which in North America is an introduced agroecosystem. It is of course of interest to synthesize such studies, but this exceeds the scope of the present analysis because it is a presentation of new results and not a review article. Studying a grazing system that is of cultural interest to Indigenous People is of scientific value and we wish to understand how these grazing systems compare to conventional grazing systems that tend to dominate the published literature.*

*In response, we added information to the Discussion section that notes the unique pasture & feedlot characteristics of the study field including new references from the literature on methane efflux.*

Since the authors compare their results to feedlot studies, it is also not clear whether the present experiment is considered as representative/comparable to a grazed pasture system or rather to a feedlot system. This should be clarified.

*The present study shares elements to both grazed pasture and feedlot systems; the animals are fed supplemental hay on a pasture in winter. We made extensive edits to the Discussion section and note that our study system shares elements to a grazed pasture and feedlot as noted.*

MINOR AND LANGUAGE COMMENTS
C) Response to prev. comment 3:
Since the authors introduce their z0 determination in detail in Section 2.4 and because the z0 values are important inputs for the footprint models, it is surly warranted that the authors present some corresponding results in the text (e.g. range of obtained z0 values with/without animals in the footprint).

*We now present median z0 values for measurements with and without bison.*

D) Response to prev. comment 9:
The rebuttal indicate that the authors principally put the use of a 'Conclusions' section into question, because their arguments could be applied to most scientific papers. I do not agree but I leave it to the editor to decide this issue.

*We feel that the end of the Discussion section adequately summarizes results and we agree that this is a personal preference, but we added a Conclusions section for completeness. We agree that it helps make the manuscript come "full circle".*

E) line 22: "...similar to eddy covariance measurements of methane efflux from a cattle feedlot during winter". It needs to be clarified here, that the mentioned feedlot results were not obtained in this study but that you mean "previously reported eddy covariance measurements ..."

*This is correct. Our results are compared against other studies that arrive at similar values to gain insight into why the methane flux measurements may be similar. We re-worded the passage.*

F) line 163: define the meaning of Fij in Eq. 4

*$F_{ij}$ is the contribution of the flux of scalar X from grid cell i,j. We changed the presentation of the equations per the Referee's suggestions in the revised manuscript, which improved the discussion of the equations because it avoids the use of 'phi_{source}', which we did not define adequately in the previous manuscript.*

G) line 321: It needs to be clarified whether the 36% is a attribution fraction or a relative uncertainty.

*Thank you for noting this, it is the contribution to total uncertainty.*

H) Figure 10 and 11A: The use of the term "probability" is strongly misleading here. The shown data rather are observed frequencies of occurrence.

*The figures show kernel density estimates, which are non-parametric estimates of probability distributions. They are probability distributions. This is not intended to be misleading and we re-worded the passage to help ensure that it is clear for the readers.*